# PULSE: Generative Phase Evolution for Non-Stationary Time Series Forecasting

Yangyou Liu [1]   Zezhi Shao [2]   Xinyu Chen [3]   Hu Chen [1]   Fei Wang [2,4]   Yuankai Wu [1]

## Abstract

Time series forecasting under non-stationarity faces a fundamental tension between capturing stable representations and adapting to distribution shifts. Existing methods implicitly rely on static historical assumptions, leading to a critical failure mode we term *Phase Amnesia*, where models become blind to the evolving global context. To resolve this, we formalize non-stationary dynamics through three physical hypotheses: wold decomposition, dynamical phase evolution, and heteroscedastic manifold generation. These principles inspire **PULSE**, a physics-informed, plug-and-play framework adopting a *Disentangle–Evolve–Simulate* design philosophy. Specifically, PULSE utilizes phase-anchored disentanglement to resolve optimization interference caused by dominant trends, employs a Phase Router to actively generate future trajectories, and introduces Statistic-Aware Mixup (SAM) to ensure robustness against out-of-distribution volatility. Empirically, PULSE enables a simple MLP backbone to achieve state-of-the-art or highly competitive performance across 12 real-world benchmarks. This validates that a correct physics-informed inductive bias is far more critical than raw architectural complexity for non-stationary forecasting. The code is available at: https://github.com/Gemost/PULSE.

## 1. Introduction

Time Series Forecasting (TSF) fundamentally addresses the challenge of conditional generation under non-stationarity (Dickey & Fuller, 1979; Liu et al., 2022b). The core tension

lies in the conflict between *learning stable representations* and *modeling distribution shifts*. To address this tension, existing deep learning approaches primarily diverge into two distinct paradigms. Normalization-based methods (Kim et al., 2021; Liu et al., 2022b; Fan et al., 2023; Liu et al., 2023b; Ye et al., 2024; Dai et al., 2024b) attempt to eliminate non-stationarity by standardizing the input data, but they always assume future statistics can be simply restored from historical moments. Conversely, structure-based methods (Wu et al., 2021; 2023; Cai et al., 2024; Dai et al., 2024a; Lin et al., 2024) aim to model the distribution shift by capturing invariant patterns like periodicity, yet they rigidly assume these structures persist into the future. Consequently, both paradigms suffer from restrictive inductive biases, leaving them incapable of expressing the complex, dynamic behaviors underlying real-world time series.

Fundamentally, these static assumptions lead to a critical failure mode we term *Phase Amnesia* (Figure 1a, left). By either aggressively neutralizing statistical shifts or strictly adhering to past cycles, existing models lose the "coordinates" of the evolving global context. Consequently, they become blind to the distinction between a genuine structural shift and mere temporary stochastic noise. To resolve Phase Amnesia, we propose a paradigm shift from conventional "passive fitting" to "generative evolution" (Figure 1a, right). Grounded in this perspective, we formalize the distinct mechanisms of non-stationary dynamics through three physical hypotheses, which inspire a novel *Disentangle–Evolve–Simulate* design philosophy for our framework:

**Hypothesis I: The Wold Decomposition Principle.** Inspired by the Wold Representation Theorem and its extension to non-stationary processes (Wold, 1938), we posit that complex time series should be conceptualized as the superposition of a *deterministic structural component* and a *stochastic residual component*. Deep forecasters often inadvertently conflate these two distinct generating mechanisms. From a classical signal processing perspective (Koopmans, 1995), this conflation leads to Optimization Interference: high-energy deterministic trends impose "spectral dominance" on the objective function, suppressing the gradient updates needed to capture low-amplitude, information-rich stochastic fluctuations.

*Constraint I:* A valid non-stationary model must explicitly

[1]College of Computer Science, Sichuan University, Chengdu, China [2]Institute of Computing Technology, Chinese Academy of Sciences, Beijing, China [3]Institute of Artificial Intelligence, University of Central Florida, Orlando, USA [4]University of Chinese Academy of Sciences, Beijing, China. Correspondence to: Yuankai Wu <wuyk0@scu.edu.cn>.

*Proceedings of the $43^{rd}$ International Conference on Machine Learning*, Seoul, South Korea. PMLR 306, 2026. Copyright 2026 by the author(s).

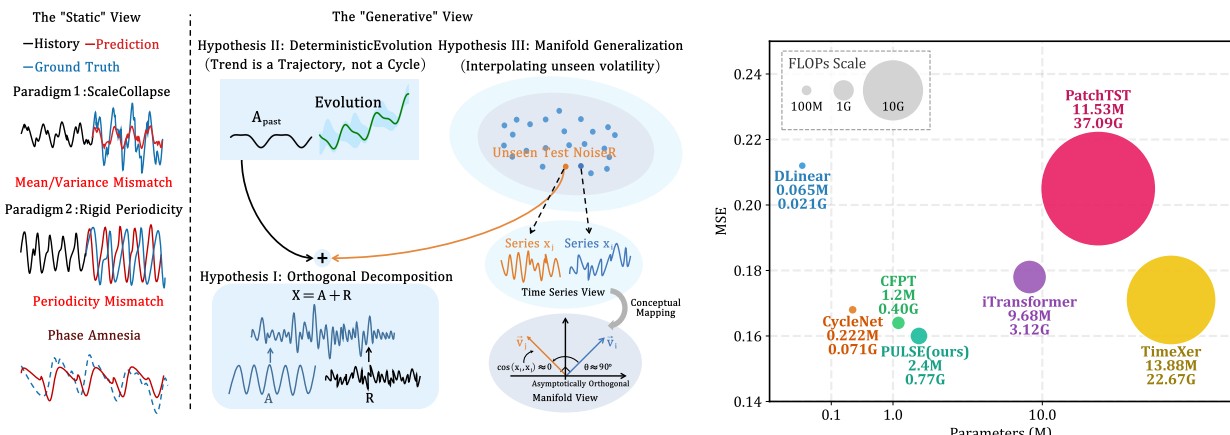

*(a)* **Motivation Analysis.** (Left) Traditional methods suffer from **Phase Amnesia** by relying on static statistics or rigid periodicity. (Right) PULSE is grounded in three physical hypotheses: (I) Wold Decomposition; (II) Dynamical Phase Evolution; (III) Heteroscedastic Residual Manifold.

*(b)* **Efficiency Analysis.** Computational efficiency of PULSE. Comparison of prediction accuracy (MSE), parameter size, and computational cost (FLOPs) on the Electricity dataset. All metrics are averaged over four prediction horizons.

*Figure 1.* Motivation and Efficiency of PULSE.

disentangle these components to optimize them in their own subspaces.

**Hypothesis II: Dynamical Phase Evolution.** In contrast to classical Fourier assumptions of global, static periodicity (Brigham, 1988), we approach time series from a nonlinear dynamical systems perspective. Real-world signals inherently exhibit non-stationary behaviors, characterized by shifting *instantaneous phase* and amplitude modulation (Huang et al., 1998). Consequently, existing models that rely on "direct historical copying" (Lin et al., 2024)—such as rigid seasonal-trend decomposition or fixed-window attention lookup—suffer from severe *extrapolation failure* when the generating mechanism shifts. History does not rigidly repeat; it evolves. We thus hypothesize that the deterministic component is not a fixed cycle, but a continuous trajectory in phase space governed by a time-varying *evolution operator* (Brunton et al., 2016).

*Constraint II:* A robust model must abandon passive historical lookup; instead, it must actively generate the future trajectory by learning this continuous evolution operator.

**Hypothesis III: Heteroscedastic Residual Manifold.** Addressing the inherent heteroscedasticity of complex systems (Bollerslev, 1986), we hypothesize that residuals do not follow a static noise profile. Instead, from an information geometry perspective (Amari, 2016), the time-varying probability distributions form a continuous statistical manifold. A fundamental limitation of standard deep forecasters is their reliance on Empirical Risk Minimization (ERM), which naively assumes independent and identically distributed (i.i.d.) noise. When test-time volatility shifts Out-of-Distribution (OOD), this i.i.d. assumption is violated, leading ERM to suffer catastrophic performance degrada-

tion (Du et al., 2021).

*Constraint III:* Residual learning must transcend passive noise fitting and be formulated as a distributional generalization task.

Based on this constrained formulation, we present **PULSE** (**P**hased **U**nfolding & **L**atent **S**tochastic **E**volution), a universal framework that constitutes a minimal realization satisfying all three hypotheses. Rather than stacking complex layers, PULSE enforces these physical constraints through: (1) **Phase-Anchored Disentanglement** to isolate orthogonal components; (2) A **Generative Phase Router** to model structural evolution; and (3) **Statistic-Aware Mixup** to simulate the residual manifold. Empirically, we demonstrate that this physics-informed disentanglement is more critical than raw model capacity.

Our main contributions are summarized as follows:

- **Physics-Informed Framework:** We formalize nonstationarity through three hypotheses and propose PULSE, a framework that resolves Phase Amnesia by disentangling evolving structures from stochastic residuals.

- **Generative Phase Evolution:** Addressing Hypothesis II, we design the Phase Router, a generative module that actively predicts the evolution of deterministic trends, breaking the static periodicity assumption.

- **Manifold-Aware Generalization:** Addressing Hypothesis III, we introduce Statistic-Aware Mixup (SAM), a theoretically grounded strategy that simulates unseen residual distributions to prevent scale collapse under heteroscedastic shifts.

- **SOTA with Minimal Complexity:** We show that our plug-and-play framework empowers a simple MLP to achieve state-of-the-art or highly competitive performance compared with complex forecasting models (Figure 1b), highlighting the importance of appropriate inductive biases.

## 2. Related Work

In recent years, deep learning-based approaches have made significant strides in addressing the challenges of non-stationary Time Series Forecasting (TSF) (Zeng et al., 2023; Nie et al., 2023; Wang et al., 2024b; Cai et al., 2024; Liu et al., 2024; Kou et al., 2025; Niu et al., 2026). To effectively model and mitigate temporal distribution shifts, existing methods generally adopt two primary strategies:

### 2.1. Normalization-based Paradigm

This family of methods aims to handle distribution shifts by mapping non-stationary series into a latent stationary space. Typically, a reversible normalization is applied before the backbone network, and predictions are inversely transformed back to the original scale. Representative works include RevIN (Kim et al., 2021), Non-stationary Transformer (Liu et al., 2022b), Dish-TS (Fan et al., 2023), SAN (Liu et al., 2023b), FAN (Ye et al., 2024), and DDN (Dai et al., 2024b). These approaches effectively stabilize training and improve generalization under smooth distribution changes. However, they usually assume that future statistical properties can be inferred from historical windows, which may not fully account for the heteroscedastic and evolving nature of real-world systems.

### 2.2. Structure-based Paradigm

Another line of work tackles non-stationarity by decomposing time series into semantic components such as trend and seasonality. *Implicit decomposition* methods embed decomposition operations into the model architecture, allowing temporal patterns to be separated during representation learning (e.g., Autoformer (Wu et al., 2021), FED-former (Zhou et al., 2022), MICN (Wang et al., 2023), DLinear (Zeng et al., 2023), TimeMixer (Wang et al., 2024a), SSCNN (Deng et al., 2024), and TimeMixer++ (Wang et al., 2025b)). In contrast, *explicit decomposition* methods introduce dedicated modules to directly extract periodic or cyclic structures before modeling (e.g., TimesNet (Wu et al., 2023), MSGNet (Cai et al., 2024), PDF (Dai et al., 2024a), CycleNet (Lin et al., 2024), TimeBase (Huang et al., 2025), TQNet (Lin et al., 2025) and TimeEmb (Xia et al., 2025)). Despite their effectiveness in capturing recurring patterns, most structure-based approaches assume relatively stable periodicity, which limits their adaptability to dynamically evolving frequency and phase shifts.

## 3. Methodology

### 3.1. Problem Formulation & Overview

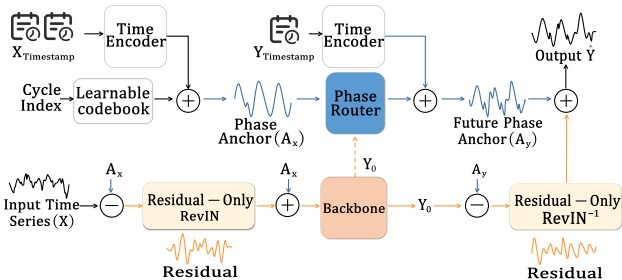

*Figure 2.* Overall architecture of the proposed **PULSE** framework.

Multivariate Time Series Forecasting (MTSF) aims to predict future sequences $\mathbf{Y} \in \mathbb{R}^{H \times C}$ over a look-ahead horizon $H$ across $C$ variates, given historical observations $\mathbf{X} \in \mathbb{R}^{T \times C}$ of look-back window $T$. Following the Generalized Decomposition Principle (Hypothesis I), we assume the underlying multivariate process is a superposition of a deterministic structural component (Phase Anchor $\mathbf{A}$) and a stochastic residual component ($\mathbf{R}$). Consequently, both the historical input and the future target are additively decomposed as: $\mathbf{X} = \mathbf{A}_x + \mathbf{R}_x, \mathbf{Y} = \mathbf{A}_y + \mathbf{R}_y$, where $\{\mathbf{A}_x, \mathbf{A}_y\}$ represent the evolving deterministic trajectory in the phase space, and $\{\mathbf{R}_x, \mathbf{R}_y\}$ capture the time-varying stochastic fluctuations. To address this, our **PULSE** framework (see Figure 2) follows a physics-informed **Disentangle–Evolve–Simulate** paradigm:

1. **Phase-Anchored Disentanglement (Section 3.2):** Decouples input into Phase Anchor $\mathbf{A}_x$ and Residual $\mathbf{R}$. We normalize *only* the residual to stabilize optimization while preserving the anchor as a structural prior.

2. **Dual-Stream Evolution (Section 3.3):** The anchor-preserved representation $\tilde{\mathbf{X}}$ flows into a generic Backbone to predict a latent future representation $\mathbf{Y}_0$, while the anchor evolves via a **Generative Phase Router** to actively generate the future reference $\mathbf{A}_y$.

3. **Generative DeNorm (Section 3.4):** Reconstruction is performed by denormalizing the residual and re-injecting $\mathbf{A}_y$, recovering the physical phase structure.

### 3.2. Phase-Anchored Disentanglement

To capture the deterministic trajectory $\mathbf{A}_x$, we introduce the **Phase-Anchor** module. Its primary function is to construct a dynamic reference frame that absorbs the high-energy structural shifts of the non-stationary series. By explicitly disentangling this dominant evolution, we can perform normalization exclusively on the stochastic residual $\mathbf{R}_x$.

**Phase Anchor Construction.** Specifically, given a global period $W$ (derived empirically via autocorrelation function (ACF) (Madsen, 2007) or pre-defined domain knowledge) and the ending timestamp $t_{\text{end}}$ of the look-back window, we map the global phase $p_W = t_{\text{end}} \bmod W$ to a learnable local codebook $\mathbf{M} \in \mathbb{R}^{L \times C}$ of size $L \times C$. The mapping resolution and the circular retrieval index $\text{idx}(h)$ for each historical step $h \in \{0, \ldots, T-1\}$ are computed as:

$$\text{idx}(h) = (p_W - h) \bmod L. \tag{1}$$

This indexing operation yields a phase-aligned discrete prototype sequence $\mathbf{M}[\text{idx}] \in \mathbb{R}^{T \times C}$. To complement these discrete prototypes with continuous global temporal cues, we incorporate standard multiscale timestamp features $\mathbf{X}_{\text{timestamp}}$ (e.g., hour-of-day, day-of-week) through a light-weight TimeEncoder. Structurally, it comprises an MLP for feature projection, a 1D convolution for local refinement, and a linear adapter. The final historical Phase-Anchor is aggregated as:

$$\mathbf{A}_x = \mathbf{M}[\text{idx}] + \text{TimeEncoder}(\mathbf{X}_{\text{timestamp}}). \tag{2}$$

**Residual-Only Normalization.** Standard instance normalization operates naively on the raw input $\mathbf{X}$. However, as posited in Hypothesis I, when the high-energy anchor dominates the signal ($\sigma_A \gg \sigma_R$), this global scaling leads to a fundamental issue we term *Variance Masking*. Dividing the entire sequence by the total standard deviation ($\sigma_X \approx \sigma_A$) inevitably suppresses the subtle residual dynamics. We formalize this "Optimization Interference" phenomenon in the following proposition:

**Proposition 3.1** (Gradient Sensitivity under Dominant Anchor). *Assume an additive decomposition* $\mathbf{X} = \mathbf{A} + \mathbf{R}$, *where the anchor component* $\mathbf{A}$ *dominates the residual* $\mathbf{R}$ *in variance, i.e.,* $\sigma_A^2 \gg \sigma_R^2$. *Then, standard normalization* $\tilde{\mathbf{X}}_{\text{std}} = \text{RevIN}(\mathbf{X})$ *scales residual gradients on the order of* $O(\sigma_A^{-1})$, *whereas our formulation* $\tilde{\mathbf{X}}_{\text{ours}} = \text{RevIN}(\mathbf{R}) + \mathbf{A}$ *scales residual gradients on the order of* $O(\sigma_R^{-1})$. *(Proof in Appendix A.1)*

Motivated by this theoretical insight, we propose to strictly decouple the normalization process to bypass the $O(\sigma_A^{-1})$ gradient attenuation. Specifically, we isolate the raw residual $\mathbf{R}_x = \mathbf{X} - \mathbf{A}_x$ and apply standard RevIN (Kim et al., 2021) *exclusively* to this stochastic component, while leaving the deterministic anchor intact:

$$\tilde{\mathbf{X}} = \underbrace{\text{RevIN}(\mathbf{R}_x)}_{\text{Standardized Residual}} + \underbrace{\mathbf{A}_x}_{\text{Deterministic Anchor}}. \tag{3}$$

By normalizing the residual via its intrinsic volatility $\sigma_R$, this design directly restores the gradient sensitivity to $O(\sigma_R^{-1})$ as guaranteed by Proposition 3.1. Consequently, it fundamentally decouples the optimization scales, allowing the network to capture low-amplitude stochastic fluctuations without sacrificing the global deterministic structure.

### 3.3. Dual-Stream Prediction

**Residual Stream (Backbone).** The anchor-preserved input $\tilde{\mathbf{X}}$ is fed into a backbone $f(\cdot)$ to predict a latent future representation $\mathbf{Y}_0 = f(\tilde{\mathbf{X}})$. While we employ a lightweight MLP as the default backbone to demonstrate that our disentanglement mechanism inherently drives state-of-the-art results, our framework is fundamentally model-agnostic. Extensive experiments (see Sec. 4) confirm its *plug-and-play* capability: integrating our framework into various mainstream forecasters (e.g., Transformers, CNNs) consistently boosts their baseline performance, validating its broad applicability across diverse architectures.

**Anchor Stream (Phase Router).** The Phase Router synthesizes the future phase anchor $\mathbf{A}_y$ from the historical anchor $\mathbf{A}_x$ and the backbone's latent sequence $\mathbf{Y}_0$. Both inputs are split into non-overlapping patches of length $P$, resulting in $N_x$ patches from $\mathbf{A}_x$ and $N_y$ patches from $\mathbf{Y}_0$. These patches are flattened and then projected via separate linear layers, which encode the *count and order* of the patches rather than the fine-grained content within each patch. This yields compact phase tokens $\mathbf{T}_x \in \mathbb{R}^{P \times d}$ and $\mathbf{T}_y \in \mathbb{R}^{P \times d}$.

A two-stage cross-attention is applied over these tokens, allowing the predictive tokens $\mathbf{T}_y$ to query and adaptively re-weight historical phase information from $\mathbf{T}_x$:

$$\mathbf{Z} = \text{Attn}(\mathbf{T}_x, \mathbf{T}_y, \mathbf{T}_y), \quad \mathbf{E} = \text{Attn}(\mathbf{T}_y, \mathbf{Z}, \mathbf{Z}). \tag{4}$$

The attended tokens $\mathbf{E}$ are projected back to the original feature dimension via an MLP, and the sequence is restored to its original order by reversing the patching operation. The final anchor is obtained by incorporating future timestamp encodings:

$$\mathbf{A}_y = \text{Proj}(\mathbf{E}) + \text{TimeEncoder}(\mathbf{Y}_{\text{timestamp}}). \tag{5}$$

This process enables the anchor to evolve dynamically, capturing shifts and modulations in future periodic patterns.

### 3.4. Generative DeNorm & Training Strategy

**Coordinate-Consistent Reconstruction.** We reconstruct the forecast $\hat{\mathbf{Y}}$ by denormalizing the extracted normalized residual and re-introducing the generated anchor:

$$\hat{\mathbf{Y}} = \text{RevIN}^{-1}(\mathbf{Y}_0 - \mathbf{A}_y) + \mathbf{A}_y. \tag{6}$$

This ensures the affine transformation acts only on fluctuations, avoiding distortion of the phase structure.

### 3.5. Training with Statistic-Aware Mixup

As posited in Hypothesis III, the residual component $\mathbf{R}$ resides on a continuous statistical manifold of time-varying

volatility. Standard empirical risk minimization on limited historical data risks overfitting to specific noise patterns, leading to failure when the residual distribution shifts OOD at test time. To bridge this gap, we formulate residual learning as a distributional generalization task.

**The Scale Collapse Trap.** While Mixup (Ansari et al., 2024) is a natural choice for manifold traversal, applying it naively to raw residual signals introduces a critical failure mode we term *Scale Collapse*. Due to the wave nature of residuals, mixing anti-phase samples triggers destructive interference, causing the latent representation scales to vanish and triggering gradient explosion. We formalize this instability in Theorem 3.2.

**Theorem 3.2** (Stability Guarantee)**.** *For any two residual signals with correlation $\rho < 1$, the naive mixup scaling $\sigma_{\text{naive}} = \sigma(\lambda \mathbf{R}_i + (1-\lambda)\mathbf{R}_j)$ systematically underestimates the feature scale and collapses to zero as $\rho \to -1$. In contrast, the proposed linear scaling $\sigma_{\text{ours}} = \lambda\sigma_i + (1-\lambda)\sigma_j$ admits a strictly positive lower bound $\min(\sigma_i, \sigma_j) > 0$, ensuring well-conditioned gradients.*

**Statistic-Aware Mixup (SAM).** Motivated by Theorem 3.2, we propose SAM to enable robust volatility simulation without numerical instability. During training, given a random permutation $\pi$ and $\lambda \sim \text{Beta}(0.15, 0.15)$, we mix $\tilde{\mathbf{X}}$ while explicitly interpolating the residual decoding statistics, so that the input-level mixup remains anchor-preserved whereas the decoding state is defined in the residual coordinate:

$$\begin{aligned}
\tilde{\mathbf{X}}^{\text{mix}} &= \lambda\tilde{\mathbf{X}} + (1-\lambda)\tilde{\mathbf{X}}^{\pi}, \\
\mu_R^{\text{mix}} &= \lambda\mu_R + (1-\lambda)\mu_R^{\pi}, \\
\sigma_R^{\text{mix}} &= \lambda\sigma_R + (1-\lambda)\sigma_R^{\pi}.
\end{aligned} \tag{7}$$

Here, $\mu_R$ and $\sigma_R$ denote the mean and standard deviation of the historical residual $\mathbf{R}_x$. The mixed input is fed into the backbone to obtain $\mathbf{Y}_0^{\text{mix}} = f(\tilde{\mathbf{X}}^{\text{mix}})$, and the Phase Router generates the corresponding future anchor $\mathbf{A}_y^{\text{mix}}$. The normalized residual part is then extracted and decoded as

$$\hat{\mathbf{Y}}^{\text{mix}} = \text{RevIN}^{-1}_{\mu_R^{\text{mix}}, \sigma_R^{\text{mix}}} \left( \mathbf{Y}_0^{\text{mix}} - \mathbf{A}_y^{\text{mix}} \right) + \mathbf{A}_y^{\text{mix}}. \tag{8}$$

*Remark* 3.3 (Residual Statistical Interpolation). SAM does not estimate the residual scale from the mixed waveform. Instead, it explicitly interpolates $(\mu_R, \sigma_R)$, preventing anti-phase residual cancellation from causing artificial scale collapse.

**Optimization Objective.** To further align with our phase-aware philosophy, we adopt the Frequency-Domain MAE (Wang et al., 2025a) as the primary optimization objective. The mixed target is constructed consistently as $\mathbf{Y}^{\text{mix}} =$ $\lambda\mathbf{Y} + (1-\lambda)\mathbf{Y}^{\pi}$, matching the same sample pairing used in SAM:

$$\mathbf{L} = \|\mathbf{F}(\hat{\mathbf{Y}}^{\text{mix}}) - \mathbf{F}(\mathbf{Y}^{\text{mix}})\|_1, \tag{9}$$

where $\mathbf{F}(\cdot)$ denotes the Fast Fourier Transform (FFT), and $\| \cdot \|_1$ refers to the $\ell_1$-norm in the frequency domain. We empirically observe that optimizing directly on the spectrum prevents the "washing out" of high-frequency components and facilitates faster convergence.

**Computational Complexity.** PULSE achieves asymptotically linear temporal complexity by operating at a fixed phase resolution $P$, making the Phase Router independent of input length. The total cost combines lightweight temporal projections and fixed-size routing attention, yielding $\mathbf{O}(T) + \mathbf{O}(P^2)$ complexity. Since $P$ is small and fixed in practice, attention behaves as a sequence-length-independent constant, so the overhead is dominated by linear projections. We further use a compact hidden dimension ($d_{\text{model}} \leq 32$) and a single-layer router. This design follows the Low-Rank Hypothesis (Huang et al., 2025): deterministic phase trajectories evolve on compact low-dimensional manifolds, making deep over-parameterized networks unnecessary. Thus, PULSE remains lightweight while modeling non-linear phase evolution. A detailed derivation is provided in Appendix B.7.

## 4. Experiment

We conduct extensive experiments to validate the effectiveness of the proposed PULSE framework, addressing three core questions:

**Q1.** Does PULSE achieve state-of-the-art forecasting accuracy with a minimal backbone?

**Q2.** As a plug-and-play framework, does it generalize across diverse backbone architectures?

**Q3.** How essential is each component, and do results align with our theoretical hypotheses?

The remainder of this section is organized as follows: Section 4.1 describes the experimental setup; Section 4.2 reports the main forecasting results and evaluates the plug-and-play generality of PULSE; and Section 4.3 presents ablation studies and further analysis.

### 4.1. Setup

**Datasets.** To ensure a comprehensive evaluation, we employ 12 widely recognized real-world datasets, stratified into long-term and short-term forecasting tasks. The long-term category comprises the ETT series (Zhou et al., 2021), Solar (Lai et al., 2018), Electricity, Traffic, and Weather datasets (Wu et al., 2021), while the short-term category encompasses the PEMS series (Liu et al., 2022a). In alignment

*Table 1.* A comparison of multivariate time series forecasting performance across 12 real-world datasets is presented below. The best results are highlighted in **bold**, the second best are underlined, and the "1$^{st}$ Count" row records the number of times each model achieves top-1 ranking. Detailed results are provided in Table 12

| Model | PULSE (ours) | | CFPT (2025) | | TimeXer (2024b) | | CycleNet (2024) | | iTransformer (2024) | | MSGNet (2024) | | TimesNet (2023) | | PatchTST (2023) | | Crossformer (2023) | | DLinear (2023) | |
|---|---|---|---|---|---|---|---|---|---|---|---|---|---|---|---|---|---|---|---|---|
| Metric | MSE | MAE | MSE | MAE | MSE | MAE | MSE | MAE | MSE | MAE | MSE | MAE | MSE | MAE | MSE | MAE | MSE | MAE | MSE | MAE |
| ETTh1 | **0.412** | **0.422** | 0.433 | 0.429 | 0.437 | 0.437 | 0.457 | 0.441 | 0.454 | 0.447 | 0.453 | 0.453 | 0.458 | 0.450 | 0.469 | 0.455 | 0.529 | 0.522 | 0.456 | 0.452 |
| ETTh2 | **0.361** | **0.391** | 0.364 | 0.393 | 0.368 | 0.396 | 0.388 | 0.409 | 0.383 | 0.407 | 0.413 | 0.427 | 0.414 | 0.427 | 0.387 | 0.407 | 0.942 | 0.684 | 0.559 | 0.515 |
| ETTm1 | **0.363** | **0.388** | 0.374 | 0.393 | 0.382 | 0.397 | 0.379 | 0.396 | 0.407 | 0.410 | 0.400 | 0.412 | 0.400 | 0.406 | 0.387 | 0.400 | 0.513 | 0.495 | 0.403 | 0.407 |
| ETTm2 | **0.262** | **0.314** | 0.269 | 0.315 | 0.274 | 0.322 | 0.266 | **0.314** | 0.288 | 0.332 | 0.289 | 0.330 | 0.291 | 0.333 | 0.281 | 0.326 | 0.757 | 0.611 | 0.350 | 0.401 |
| Electricity | **0.160** | **0.255** | 0.164 | 0.259 | 0.171 | 0.270 | 0.168 | 0.259 | 0.178 | 0.270 | 0.194 | 0.301 | 0.193 | 0.295 | 0.205 | 0.290 | 0.244 | 0.334 | 0.212 | 0.300 |
| Solar | **0.194** | **0.243** | 0.233 | 0.267 | 0.237 | 0.302 | 0.210 | 0.261 | 0.233 | 0.262 | 0.263 | 0.292 | 0.301 | 0.319 | 0.270 | 0.307 | 0.641 | 0.639 | 0.330 | 0.401 |
| Traffic | 0.460 | 0.286 | 0.470 | 0.289 | 0.466 | 0.287 | 0.472 | 0.301 | **0.428** | **0.282** | 0.660 | 0.382 | 0.620 | 0.336 | 0.481 | 0.304 | 0.550 | 0.304 | 0.625 | 0.383 |
| Weather | **0.239** | 0.271 | 0.240 | **0.267** | 0.241 | 0.271 | 0.243 | 0.271 | 0.258 | 0.278 | 0.249 | 0.278 | 0.259 | 0.287 | 0.259 | 0.281 | 0.259 | 0.315 | 0.265 | 0.317 |
| PEMS03 | **0.103** | **0.209** | 0.143 | 0.250 | 0.112 | 0.214 | 0.118 | 0.226 | 0.113 | 0.221 | 0.150 | 0.251 | 0.147 | 0.248 | 0.180 | 0.291 | 0.169 | 0.282 | 0.278 | 0.375 |
| PEMS04 | **0.104** | 0.210 | 0.158 | 0.264 | 0.105 | **0.209** | 0.119 | 0.232 | 0.111 | 0.221 | 0.122 | 0.239 | 0.129 | 0.241 | 0.195 | 0.307 | 0.209 | 0.314 | 0.295 | 0.388 |
| PEMS07 | 0.094 | 0.190 | 0.125 | 0.226 | **0.085** | **0.182** | 0.113 | 0.214 | 0.101 | 0.204 | 0.122 | 0.227 | 0.125 | 0.226 | 0.211 | 0.303 | 0.235 | 0.315 | 0.329 | 0.396 |
| PEMS08 | **0.138** | **0.220** | 0.187 | 0.266 | 0.175 | 0.250 | 0.150 | 0.246 | 0.150 | 0.226 | 0.205 | 0.285 | 0.193 | 0.271 | 0.280 | 0.321 | 0.268 | 0.307 | 0.379 | 0.416 |
| 1$^{st}$ Count | **10** | **8** | 0 | 1 | 1 | 2 | 0 | 1 | 1 | 1 | 0 | 0 | 0 | 0 | 0 | 0 | 0 | 0 | 0 | 0 |

with standard evaluation protocols, the historical look-back window $T$ is fixed to 96 for all tasks. The prediction horizon $H$ is configured as $\{12, 24, 48, 96\}$ for the PEMS datasets and $\{96, 192, 336, 720\}$ for the long-term benchmarks. Detailed specifications are provided in Appendix B.1.

**Baselines.** To rigorously evaluate the performance of PULSE, we compare it against a wide range of state-of-the-art forecasting models, including : CFPT (Kou et al., 2025), TimeXer (Wang et al., 2024b), CycleNet (Lin et al., 2024), iTransformer (Liu et al., 2024), MSGNet (Cai et al., 2024), TimesNet (Wu et al., 2023), PatchTST (Nie et al., 2023), Crossformer (Zhang & Yan, 2023), and DLinear (Zeng et al., 2023).

### 4.2. Main Results

**Forecasting Performance.** Table 1 compares PULSE with state-of-the-art baselines across twelve real-world datasets. PULSE ranks first on 10 out of 12 datasets in terms of MSE and on 8 out of 12 datasets in terms of MAE, yielding 18 first-place entries among the 24 reported results. It also substantially outperforms CFPT, a recent long-term forecasting baseline based on cross-frequency interactions, indicating that phase-anchored disentanglement captures non-stationary evolution more effectively than conventional frequency-based mixing.

The cases where PULSE does not rank first mainly occur on highly dense multivariate benchmarks, such as Traffic with 862 variables and PEMS07 with 883 variables. On these datasets, attention-based models such as iTransformer and

TimeXer remain competitive because their inverted attention mechanisms explicitly model cross-channel correlations. This inductive bias is useful when forecasting errors are influenced by both temporal non-stationarity and sensor-level dependencies. By contrast, PULSE is designed to model temporal phase evolution with a minimal backbone, rather than to explicitly encode inter-variable graphs. This design choice explains the slightly lower rank of PULSE on Traffic and PEMS07, where dense cross-variate interactions play a more prominent role. Nevertheless, PULSE still achieves the best performance on PEMS08 with 170 variables and remains among the strongest methods on Traffic. These results suggest that phase-evolution modeling is highly effective when temporal distribution shifts dominate, while incorporating graph-aware or channel-aware mechanisms is a promising direction for densely correlated multivariate benchmarks.

**Plug-and-play Capability.** To demonstrate the versatility of our framework, Table 2 evaluates PULSE by replacing its default lightweight MLP backbone with four state-of-the-art architectures. This reveals PULSE's strong universality, as it consistently unlocks further performance gains across diverse model families. Most notably, DLinear achieves a striking **33.3%** improvement on ETTh2, effectively alleviating the "static scale" limitation inherent in linear forecasters. Advanced architectures such as iTransformer and TimesNet also show robust synergistic gains within our framework.

However, a performance drop of **-17.5%** occurs when PatchTST is employed as the backbone on the Traffic dataset. This exception requires a more careful interpretation, be-

*Table 2.* Evaluation of plug-and-play capability. "Promotion" denotes the relative improvement, highlighting the ability of PULSE to enhance different model architectures. Detailed results are provided in Table 13.

| Model | ETTh1 | | ETTh2 | | ETTm1 | | ETTm2 | | ECL | | Traffic | | Weather | | Solar | |
|---|---|---|---|---|---|---|---|---|---|---|---|---|---|---|---|---|
| | MSE | MAE | MSE | MAE | MSE | MAE | MSE | MAE | MSE | MAE | MSE | MAE | MSE | MAE | MSE | MAE |
| iTransformer | 0.454 | 0.447 | 0.383 | 0.407 | 0.407 | 0.410 | 0.288 | 0.332 | 0.178 | 0.270 | 0.428 | 0.282 | 0.258 | 0.282 | 0.233 | 0.262 |
| **+ PULSE** | **0.409** | **0.422** | **0.372** | **0.399** | **0.362** | **0.383** | **0.269** | **0.316** | **0.159** | **0.258** | **0.424** | **0.272** | **0.237** | **0.275** | **0.193** | **0.257** |
| Promotion | **9.9%** | **5.6%** | **2.9%** | **2.0%** | **11.1%** | **6.6%** | **6.5%** | **4.8%** | **10.7%** | **4.4%** | **0.9%** | **3.5%** | **8.1%** | **2.5%** | **17.2%** | **1.9%** |
| PatchTST | 0.469 | 0.454 | 0.387 | 0.407 | 0.387 | 0.400 | 0.281 | 0.326 | 0.205 | 0.290 | 0.481 | 0.304 | 0.259 | 0.281 | 0.270 | 0.307 |
| **+ PULSE** | **0.440** | **0.448** | **0.375** | **0.403** | **0.367** | **0.396** | **0.269** | **0.319** | **0.176** | **0.279** | 0.565 | **0.301** | **0.242** | **0.274** | **0.216** | **0.280** |
| Promotion | **6.2%** | **1.3%** | **3.1%** | **1.0%** | **5.2%** | **1.0%** | **4.3%** | **2.1%** | **14.4%** | **3.8%** | **-17.5%** | **1.0%** | **6.6%** | **2.5%** | **20.0%** | **8.8%** |
| TimesNet | 0.458 | 0.507 | 0.414 | 0.427 | 0.400 | 0.406 | 0.291 | 0.333 | 0.192 | 0.295 | 0.620 | 0.336 | 0.259 | 0.287 | 0.301 | 0.319 |
| **+ PULSE** | **0.422** | **0.429** | **0.383** | **0.409** | **0.372** | **0.397** | **0.269** | **0.318** | **0.182** | **0.287** | **0.553** | **0.319** | **0.234** | **0.285** | **0.203** | **0.266** |
| Promotion | **7.9%** | **15.4%** | **7.5%** | **4.2%** | **7.0%** | **2.2%** | **7.6%** | **4.5%** | **5.2%** | **2.7%** | **10.8%** | **5.1%** | **9.7%** | **0.6%** | **32.6%** | **16.6%** |
| DLinear | 0.456 | 0.452 | 0.559 | 0.515 | 0.403 | 0.407 | 0.350 | 0.401 | 0.212 | 0.300 | 0.625 | 0.383 | 0.265 | 0.317 | 0.330 | 0.401 |
| **+ PULSE** | **0.428** | **0.424** | **0.373** | **0.396** | **0.376** | **0.394** | **0.274** | **0.322** | **0.168** | **0.267** | **0.515** | **0.313** | **0.250** | **0.285** | **0.216** | **0.272** |
| Promotion | **6.1%** | **6.2%** | **33.3%** | **23.1%** | **6.7%** | **3.2%** | **21.7%** | **19.7%** | **20.8%** | **11.0%** | **17.6%** | **18.3%** | **5.7%** | **10.1%** | **34.5%** | **32.2%** |

*Table 3.* Ablation studies of the proposed PULSE framework.

| Model | ETTh1 | | ETTh2 | | ETTm1 | | ETTm2 | | ECL | | Traffic | | Weather | | Solar | | Average | | Promotion | |
|---|---|---|---|---|---|---|---|---|---|---|---|---|---|---|---|---|---|---|---|---|
| | MSE | MAE | MSE | MAE | MSE | MAE | MSE | MAE | MSE | MAE | MSE | MAE | MSE | MAE | MSE | MAE | MSE | MAE | MSE | MAE |
| **PULSE (Ours)** | **0.412** | **0.422** | **0.361** | **0.391** | **0.363** | **0.388** | **0.262** | **0.314** | **0.160** | **0.255** | **0.460** | 0.286 | **0.239** | **0.271** | **0.194** | **0.243** | **0.306** | **0.321** | - | - |
| w/o Phase Anchor | 0.430 | 0.425 | 0.365 | 0.393 | 0.375 | 0.393 | 0.273 | 0.319 | 0.173 | 0.264 | 0.497 | 0.309 | 0.252 | 0.283 | 0.239 | 0.289 | 0.326 | 0.334 | 6.0% | 4.0% |
| w/o SAM | 0.417 | 0.424 | 0.366 | 0.393 | 0.364 | 0.389 | 0.265 | 0.316 | **0.160** | 0.256 | 0.498 | **0.282** | 0.241 | **0.271** | 0.203 | 0.249 | 0.314 | 0.323 | 2.6% | 0.6% |
| w/o Statistic-Aware | 0.419 | 0.425 | 0.367 | 0.393 | 0.372 | 0.391 | 0.268 | 0.317 | **0.160** | 0.257 | 0.475 | 0.292 | 0.243 | 0.275 | 0.199 | 0.249 | 0.313 | 0.325 | 2.2% | 1.2% |
| w/o Phase Router | 0.426 | 0.426 | 0.369 | 0.396 | 0.373 | 0.395 | 0.268 | 0.318 | 0.163 | 0.258 | 0.480 | 0.291 | 0.244 | 0.275 | 0.200 | 0.248 | 0.315 | 0.326 | 2.9% | 1.5% |

cause Traffic is a densely correlated multivariate benchmark where channel-independent designs are often viewed as a potential weakness. Yet channel independence alone cannot explain the degradation, since the default MLP backbone in PULSE is also channel-independent and still performs competitively on Traffic. Instead, the limitation more likely stems from an architectural mismatch between PatchTST's patch-level aggregation and the residual stream produced by Phase-Anchored Disentanglement. After the dominant structural component is absorbed by the anchor stream, the remaining residual becomes more stochastic and less locally smooth, weakening the stable local-patch assumption used by PatchTST. This issue is further amplified on Traffic, where dense sensor interactions make local temporal patterns more complex. In contrast, the lightweight MLP imposes a weaker locality prior and can model the residual signal more directly under the SAM-based training objective. Therefore, the degradation of PatchTST+PULSE on Traffic is better interpreted as a dataset-specific inductive-bias mismatch, rather than as an inherent limitation of PULSE or channel-independent backbones. We further dissect the contribution of each component in the following ablation studies.

## 4.3. Ablation Studies and Analysis

**Ablation Studies.** To validate the Disentangle–Evolve–Simulate design, we conduct component-wise ablation studies across all evaluated datasets. PULSE consists of three synergistic mechanisms, each addressing a distinct aspect of non-stationary forecasting: (1) removing the **Phase Anchor** eliminates the structural reference frame, forcing the model to learn non-stationary evolution from entangled raw observations; (2) removing the **Phase Router** reduces the framework to a static architecture that handles different temporal dynamics uniformly, without adapting the future phase trajectory; (3) removing **SAM** or disabling the **Statistic-Aware** mechanism removes statistical calibration, increasing vulnerability to OOD shifts and unstable residual scaling.

As shown in Table 3, PULSE consistently outperforms all variants, confirming the necessity of each module. The **Phase Anchor** is the foundational component; removing it causes the largest degradation, with an average MSE drop of **6.0%**. This result supports Hypothesis I, showing that a sta-

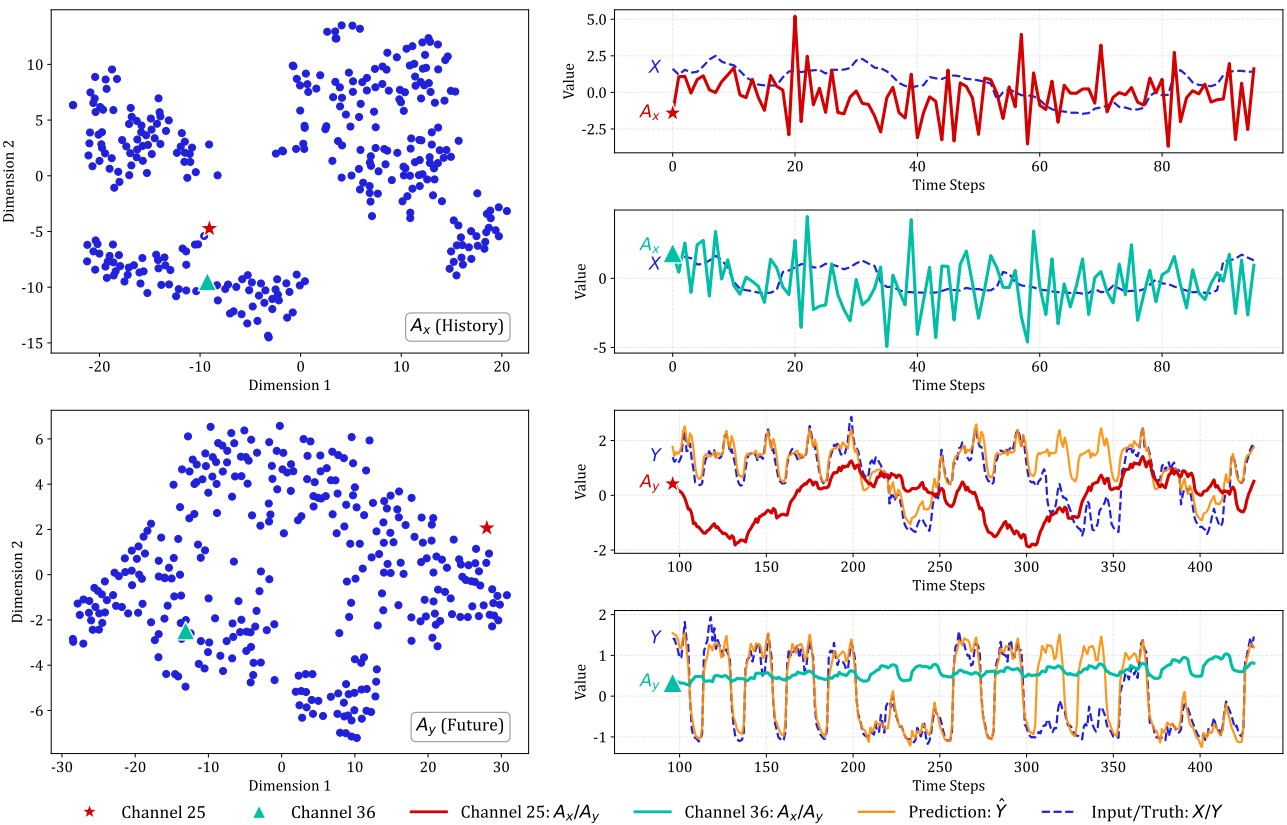

*Figure 3.* Visual analysis of learned Phase Anchors versus raw data. The data are sourced from the Electricity dataset.

ble disentanglement basis is necessary for forecasting under non-stationarity. The **Phase Router** provides a **2.9%** gain by dynamically dispatching temporal patterns, confirming that future structures should be generated rather than copied from history. Finally, **SAM** and the **Statistic-Aware** mechanism yield gains of 2.6% and 2.2%, respectively. These results support Theorem 3.2, showing that statistical interpolation helps prevent scale collapse and mitigate volatility shifts. The advantage of PULSE does not come from simply accumulating modules, but from their coherent integration: the Anchor provides the structural basis, the Router enables adaptive evolution, and SAM improves robustness under distributional changes.

**Analysis of Dynamical Phase Evolution.** To validate Hypothesis II and demonstrate the efficacy of the Generative Phase Router, we inspect the intrinsic representations learned by PULSE. Figure 3 visualizes the t-SNE (Maaten & Hinton, 2008) embeddings of the deterministic Phase Anchors, including the History Anchor ($\mathbf{A}_x$) and the generated Future Anchor ($\mathbf{A}_y$), together with their raw temporal sequences.

The visualization highlights the one-to-many mapping problem in non-stationary forecasting: although Channels 25 and

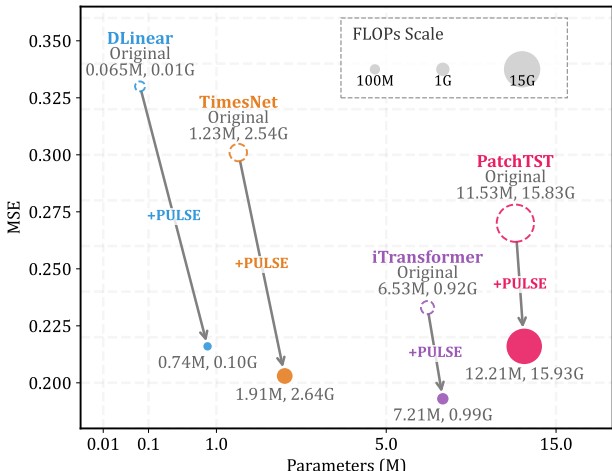

*Figure 4.* **Plug-and-Play Efficiency of PULSE on the Solar Dataset.** We illustrate the performance shift of four backbone models after being equipped with PULSE. The comparison covers MSE, Parameters, and FLOPs, averaged over prediction horizons $H \in \{96, 192, 336, 720\}$. Arrows indicate clear accuracy improvements with modest additional computational overhead.

36 share similar historical patterns and cluster closely in $\mathbf{A}_x$, their future trajectories diverge markedly. The Generative

*Table 4.* **Plug-and-play accuracy–efficiency trade-off.** The look-back length is fixed to 96, and all results are averaged over prediction horizons $H \in \{96, 192, 336, 720\}$. Runtime efficiency is measured by repeatedly processing a single batch 100 times; Lat. denotes the median inference latency in milliseconds, and Mem. denotes the peak GPU memory usage in GB.

| Model | ETTh1 | | | | Electricity | | | | Weather | | | | Solar | | | |
|---|---|---|---|---|---|---|---|---|---|---|---|---|---|---|---|---|
| | MSE | MAE | Lat.(ms) | Mem.(GB) | MSE | MAE | Lat.(ms) | Mem.(GB) | MSE | MAE | Lat.(ms) | Mem.(GB) | MSE | MAE | Lat.(ms) | Mem.(GB) |
| iTransformer | 0.454 | 0.447 | 4.068 | 0.130 | 0.178 | 0.270 | 20.910 | 0.904 | 0.258 | 0.282 | 5.029 | 0.238 | 0.233 | 0.262 | 8.358 | 0.291 |
| **+ PULSE** | **0.409** | **0.422** | 6.598 | 0.131 | **0.159** | **0.258** | 28.580 | 0.945 | **0.237** | **0.275** | 8.965 | 0.239 | **0.193** | **0.257** | 11.178 | 0.316 |
| PatchTST | 0.469 | 0.454 | 8.028 | 0.415 | 0.205 | 0.290 | 87.483 | 4.315 | 0.259 | 0.281 | 22.023 | 1.163 | 0.270 | 0.307 | 38.678 | 1.868 |
| **+ PULSE** | **0.440** | **0.448** | 9.908 | 0.422 | **0.176** | **0.279** | 94.188 | 4.364 | **0.242** | **0.274** | 25.655 | 1.175 | **0.216** | **0.280** | 40.290 | 1.889 |
| TimesNet | 0.458 | 0.507 | 40.848 | 0.993 | 0.192 | 0.295 | 9.843 | 0.312 | 0.259 | 0.287 | 39.980 | 0.999 | 0.301 | 0.319 | 10.138 | 0.293 |
| **+ PULSE** | **0.422** | **0.429** | 43.750 | 0.999 | **0.182** | **0.287** | 13.888 | 0.447 | **0.234** | **0.285** | 43.158 | 1.013 | **0.203** | **0.266** | 13.040 | 0.316 |
| DLinear | 0.456 | 0.452 | 0.438 | 0.043 | 0.212 | 0.300 | 0.625 | 0.158 | 0.265 | 0.317 | 0.485 | 0.064 | 0.330 | 0.401 | 0.490 | 0.086 |
| **+ PULSE** | **0.428** | **0.424** | 3.828 | 0.077 | **0.168** | **0.267** | 8.010 | 0.409 | **0.250** | **0.285** | 4.615 | 0.164 | **0.216** | **0.272** | 5.490 | 0.210 |

Phase Router resolves this ambiguity by mapping proximate historical anchors to distinct regions in $\mathbf{A}_y$, enabling future-dependent phase evolution rather than direct historical copying. Furthermore, the temporal sequences reveal a clear structural transition: while $\mathbf{A}_x$ retains high-frequency variations to represent local non-stationary shocks, $\mathbf{A}_y$ evolves into a smoother and lower-rank trajectory that serves as a future-oriented structural reference. This transition suggests that PULSE actively generates the deterministic phase trajectory while separating it from transient stochastic fluctuations. Together, these observations support Hypothesis II by showing that PULSE learns future-oriented phase evolution instead of relying on static historical patterns. We provide a detailed spectral analysis of this phenomenon in Appendix B.6.

**Plug-and-Play Efficiency Analysis.** As shown in Figure 4 and Table 4, integrating PULSE into existing architectures yields substantial error reductions with modest overhead, indicating that the correction module improves accuracy without dominating the cost of the host model. On Solar, DLinear improves from 0.330 to 0.216 MSE, with FLOPs increasing only from 0.009G to 0.105G, turning a simple linear model into a competitive forecaster. Notably, the gains also extend to stronger backbones: TimesNet improves from 0.301 to 0.203 MSE, and iTransformer improves from 0.233 to 0.193 MSE, without changing their fundamental complexity class.

Beyond FLOPs, Table 4 reports actual inference latency and peak GPU memory, providing a more direct assessment of deployment cost. For medium-to-heavy backbones, PULSE usually introduces only a few milliseconds of additional latency and minor memory overhead. For example, on Solar, PatchTST improves from 0.270 to 0.216 MSE, while latency increases slightly from 38.678 ms to 40.290 ms and memory usage increases from 1.868 GB to 1.889 GB. For DLinear, the relative latency increase is larger because

the baseline cost is nearly zero, but the absolute overhead remains small compared with the accuracy gain. Overall, PULSE serves as a practical plug-and-play accuracy booster, improving robustness to non-stationary shifts while maintaining acceptable runtime and memory costs across diverse backbone scales.

## 5. Conclusion

This paper introduced PULSE, a physics-informed framework that shifts non-stationary forecasting from passive fitting to generative evolution. By integrating Phase-Anchored Disentanglement and a Generative Phase Router, PULSE effectively addresses Phase Amnesia and enables accurate synthesis of future trajectories. Extensive experiments across 12 real-world datasets show that PULSE empowers a lightweight MLP backbone to achieve state-of-the-art performance on most benchmarks, while maintaining linear-level computational efficiency and remaining competitive with complex Transformer-based models. Its plug-and-play nature and strong balance of accuracy, efficiency, and robustness position PULSE as a promising foundation for challenging real-world forecasting tasks.

## Acknowledgements

This work was supported in part by the National Natural Science Foundation of China under Grant No. 62406206, by the Fundamental Research Funds for the Central Universities, by the National Natural Science Foundation of China under Grant U25A20439, by the Sichuan Provincial Natural Science Foundation under Grant No. 2026NSFSC0426, by the NSFC under Grant Nos. 62372430 and 62502505, by the Youth Innovation Promotion Association CAS under No. 2023112, by the Postdoctoral Fellowship Program of CPSF under Grant Number GZC20251078, and by the China Postdoctoral Science Foundation under Grant No. 2025M77154.

## Impact Statement

This paper aims to advance the field of Time Series Forecasting by addressing non-stationarity with high efficiency. While our framework has potential applications in domains such as energy and transportation, we do not foresee any specific negative societal consequences or ethical concerns that require highlighting here.

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

# A. Theoretical Analysis and Proofs

## A.1. Proof of Gradient Sensitivity (Proposition 3.1)

**Proposition (Gradient Sensitivity under Dominant Anchor).** *Assume an additive decomposition $\mathbf{x} = \mathbf{A} + \mathbf{R}$, where the anchor component $\mathbf{A}$ dominates the residual $\mathbf{R}$ in variance, i.e., $\sigma_A^2 \gg \sigma_R^2$. Then, standard normalization $\tilde{\mathbf{x}}_{\text{std}} = \text{RevIN}(\mathbf{x})$ scales residual gradients on the order of $O(\sigma_A^{-1})$, whereas our formulation $\tilde{\mathbf{x}}_{\text{ours}} = \text{RevIN}(\mathbf{R}) + \mathbf{A}$ scales residual gradients on the order of $O(\sigma_R^{-1})$.*

*Proof.* RevIN normalizes an input sequence $\mathbf{x} \in \mathbb{R}^T$ by scaling it with its standard deviation:

$$\text{RevIN}(\mathbf{x}) = \frac{\mathbf{x} - \mu(\mathbf{x})}{\sigma(\mathbf{x})},$$

where the mean $\mu(\mathbf{x}) = \frac{1}{T}\sum_{i=1}^{T} x_i$ and the variance is defined as

$$\sigma^2(\mathbf{x}) = \frac{1}{T}\sum_{i=1}^{T}(x_i - \mu(\mathbf{x}))^2.$$

Assume an additive decomposition $\mathbf{x} = \mathbf{A} + \mathbf{R}$, with negligible covariance between $\mathbf{A}$ and $\mathbf{R}$, and that $\mathbf{A}$ dominates in variance ($\sigma_A^2 \gg \sigma_R^2$). The total variance can be expanded as:

$$\sigma^2(\mathbf{x}) = \sigma_A^2 + \sigma_R^2 + 2\text{Cov}(\mathbf{A}, \mathbf{R}) \approx \sigma_A^2\left(1 + \frac{\sigma_R^2}{\sigma_A^2}\right).$$

This implies that the standard deviation is dominated by the anchor: $\sigma(\mathbf{x}) = \Theta(\sigma_A)$.

The Jacobian of RevIN with respect to $\mathbf{R}$ admits the standard instance-normalization decomposition:

$$\frac{\partial \text{RevIN}(\mathbf{x})}{\partial \mathbf{R}} = \frac{1}{\sigma(\mathbf{x})}\left(\mathbf{I} - \frac{1}{T}\mathbf{1}\mathbf{1}^\top\right) - \frac{1}{T\sigma(\mathbf{x})^3}(\mathbf{x} - \mu_x\mathbf{1})(\mathbf{x} - \mu_x\mathbf{1})^\top.$$

The first term dominates with a spectral norm of $\Theta(\sigma(\mathbf{x})^{-1}) = \Theta(\sigma_A^{-1})$, while the second term is a rank-one correction of strictly lower order. Therefore, the gradient magnitude scales as:

$$\left\|\frac{\partial \text{RevIN}(\mathbf{x})}{\partial \mathbf{R}}\right\|_2 = \Theta\left(\frac{1}{\sigma_A}\right).$$

In contrast, our method applies normalization directly to the residual:

$$\tilde{\mathbf{x}}_{\text{ours}} = g(\mathbf{R}) + \mathbf{A}, \qquad g(\mathbf{R}) = \frac{\mathbf{R} - \mu(\mathbf{R})}{\sigma(\mathbf{R})}.$$

Since $\mathbf{A}$ is independent of $\mathbf{R}$ in this formulation,

$$\frac{\partial \tilde{\mathbf{x}}_{\text{ours}}}{\partial \mathbf{R}} = \frac{\partial g(\mathbf{R})}{\partial \mathbf{R}}.$$

The Jacobian of $g$ admits an analogous decomposition, where the scaling factor depends on the intrinsic volatility of the residual $\sigma_R$. Hence,

$$\left\|\frac{\partial \tilde{\mathbf{x}}_{\text{ours}}}{\partial \mathbf{R}}\right\|_2 = \Theta\left(\frac{1}{\sigma_R}\right).$$

Combining the above results yields the ratio of gradient scales:

$$\frac{\|\partial \tilde{\mathbf{x}}_{\text{ours}}/\partial \mathbf{R}\|_2}{\|\partial \tilde{\mathbf{x}}_{\text{std}}/\partial \mathbf{R}\|_2} = \Theta\left(\frac{\sigma_A}{\sigma_R}\right) \gg 1,$$

which proves that our formulation prevents gradient attenuation caused by the dominant anchor. $\square$

## A.2. Proof of Scale Collapse (Theorem 3.2)

**Theorem A.1** (Stability Guarantee). *For any two residual signals with correlation $\rho < 1$, the naive mixup scaling $\sigma_{\text{naive}} = \sigma(\lambda\mathbf{R}_i + (1-\lambda)\mathbf{R}_j)$ systematically underestimates the feature scale and collapses to zero as $\rho \to -1$. In contrast, the proposed linear scaling $\sigma_{\text{ours}} = \lambda\sigma_i + (1-\lambda)\sigma_j$ admits a strictly positive lower bound $\min(\sigma_i, \sigma_j) > 0$, ensuring well-conditioned gradients.*

*Proof.* Consider two normalized signals $\mathbf{R}_i$ and $\mathbf{R}_j$ with standard deviations $\sigma_i$ and $\sigma_j$, and correlation coefficient $\rho$. The variance of their linear mixup is given by

$$\sigma_{\text{naive}}^2 = \lambda^2\sigma_i^2 + (1-\lambda)^2\sigma_j^2 + 2\lambda(1-\lambda)\rho\sigma_i\sigma_j.$$

By contrast, our method defines the scale as a convex combination

$$\sigma_{\text{ours}} = \lambda\sigma_i + (1-\lambda)\sigma_j.$$

Dividing the two expressions yields

$$\left(\frac{\sigma_{\text{naive}}}{\sigma_{\text{ours}}}\right)^2 = 1 - \frac{2\lambda(1-\lambda)\sigma_i\sigma_j(1-\rho)}{(\lambda\sigma_i + (1-\lambda)\sigma_j)^2}.$$

becomes exactly $1 - 1 = 0$ when $\rho = -1$ and $\lambda = \frac{\sigma_j}{\sigma_i + \sigma_j}$.

If such a vanishing scale is used in decoding,

$$\hat{\mathbf{y}} = \mathbf{z} \cdot \sigma_{\text{naive}},$$

any non-zero target $\mathbf{y}_{\text{mix}}$ forces the latent representation $\mathbf{z}$ to diverge, resulting in numerical instability and gradient explosion.

In contrast, since

$$\sigma_{\text{ours}} \geq \min(\sigma_i, \sigma_j) > 0,$$

our method prevents scale collapse and ensures well-conditioned latent representations even under adversarial phase alignment. $\square$

*Table 5.* Dataset Statistics. Var is the number of variables, Length is the dataset length, Freq is the sampling frequency, $W$ is the global period, $T$ is the length of look-back window, and $H$ is the forecasting horizon.

| Dataset | Vars. | Length | Freq | $W$ | $T$ | $H$ |
|---|---|---|---|---|---|---|
| ETTh1 | 7 | 14,400 | 1 hour | 24 | 96 | 96–720 |
| ETTh2 | 7 | 14,400 | 1 hour | 24 | 96 | 96–720 |
| ETTm1 | 7 | 57,600 | 15 min | 96 | 96 | 96–720 |
| ETTm2 | 7 | 57,600 | 15 min | 96 | 96 | 96–720 |
| Electricity | 321 | 26,304 | 1 hour | 168 | 96 | 96–720 |
| Solar | 137 | 52,560 | 10 min | 144 | 96 | 96–720 |
| Traffic | 862 | 17,544 | 1 hour | 168 | 96 | 96–720 |
| Weather | 21 | 52,696 | 10 min | 144 | 96 | 96–720 |
| PEMS03 | 358 | 26,208 | 5 min | 288 | 96 | 12–96 |
| PEMS04 | 307 | 16,992 | 5 min | 288 | 96 | 12–96 |
| PEMS07 | 883 | 28,224 | 5 min | 288 | 96 | 12–96 |
| PEMS08 | 170 | 17,856 | 5 min | 288 | 96 | 12–96 |

# B. More Details of PULSE

## B.1. Datasets

To ensure a comprehensive evaluation of the proposed framework, we conduct experiments on 12 widely recognized real-world datasets, covering both long-term and short-term forecasting tasks. The detailed statistics, including variable counts, sampling frequencies, and prediction settings, are summarized in Table 5.

- **ETT Series** (Zhou et al., 2021): This dataset records electricity transformer signals from July 2016 to July 2018, including oil temperature and power load features. It consists of four subsets: ETTh1/ETTh2 with 1-hour intervals and ETTm1/ETTm2 with 15-minute intervals.

- **Electricity** (Wu et al., 2021): This dataset records hourly electricity consumption from 321 clients between 2012 and 2014.

- **Solar-Energy** (Lai et al., 2018): This dataset contains solar power production from 137 PV plants in 2006, sampled every 10 minutes.

- **Traffic** (Wu et al., 2021): This dataset records hourly road occupancy rates from 862 sensors on San Francisco Bay area freeways from January 2015 to December 2016.

- **Weather** (Wu et al., 2021): This dataset includes 21 meteorological indicators collected every 10 minutes from the Max Planck Biogeochemistry weather station in 2020.

- **PEMS Series**: This dataset contains California traffic network data collected at 5-minute intervals. Following SCINet (Liu et al., 2022a), we use four public subsets: PEMS03, PEMS04, PEMS07, and PEMS08.

## B.2. Experiment Details

The proposed model is implemented in PyTorch (Paszke et al., 2019) and executed on a single NVIDIA GeForce RTX 4090 GPU (24 GB). Training is performed using the Adam (Kingma & Ba, 2014) optimizer with a fixed learning rate of 0.005 and an early stopping patience of 5 epochs, with a total budget of 30 epochs. To ensure reproducibility, all experiments are conducted with a fixed random seed of 2024. Following established benchmarks like iTransformer (Liu et al., 2024) and TimesNet (Wu et al., 2023), the training-validation-test split ratios are set to 6:2:2 for the ETT and PEMS series, and 7:1:2 for the remaining datasets.

Regarding the architecture, the $d_{model}$ of the MLP backbone is fixed at 512, while the $d_{model}$ for auxiliary components, such as the phase router, is set to 16 or 32 depending on the task. A consistent dropout rate of 0.1 is applied across all layers. To optimize hardware utilization and convergence, the batch size is set according to dataset scale: 256 for ETT and Weather; 64 for Electricity, Traffic, and Solar; and 32 for the PEMS series.

Regarding key hyperparameters, the global period $W$ for each dataset is empirically determined by the autocorrelation function (ACF) (Madsen, 2007), with values reported in Table 5. The codebook size is selected from $\{12, 24, 32, 48, 96\}$ in most cases, except for Weather at horizon 336, where it is set to 72. To capture multi-scale temporal dynamics, we employ task-specific timestamp features from {MinuteOfHour, HourOfDay, DayOfWeek, DayOfMonth, DayOfYear}. The specific combination depends on each dataset's sampling frequency and inherent periodicity. For instance, we use HourOfDay for the hourly-sampled ETTh1 dataset with a daily cycle, while additionally incorporating DayOfWeek for the Traffic dataset to capture weekly seasonality. For the router module, the patch length is tuned within $\{4, 6, 8, 12, 24\}$, and the number of attention heads is fixed at 1 to maintain computational efficiency.

## B.3. Quantifying Phase Amnesia

To further substantiate the Phase Amnesia failure mode, we quantify the discrepancy between the historical input and the future target. Given an input sequence $\mathbf{X}$ and its future target $\mathbf{Y}$, we evaluate three complementary forms of mismatch: mean shift (MS), standard-deviation shift (SS), and spectral mismatch (SM):

$$
\begin{aligned}
\mathrm{MS} &= \frac{|\mu_{\mathbf{Y}} - \mu_{\mathbf{X}}|}{|\mu_{\mathbf{Y}}| + |\mu_{\mathbf{X}}| + \epsilon}, \\
\mathrm{SS} &= \frac{|\sigma_{\mathbf{Y}} - \sigma_{\mathbf{X}}|}{\sigma_{\mathbf{Y}} + \sigma_{\mathbf{X}} + \epsilon}, \\
\mathrm{SM} &= \frac{1}{2} \sum_f \left| \tilde{\mathbf{A}}_{\mathbf{X}}(f) - \tilde{\mathbf{A}}_{\mathbf{Y}}(f) \right|.
\end{aligned}
\tag{10}
$$

*Table 6.* **Quantification of history–future mismatch under** $T = 96$. MS, SS, and SM denote mean shift, standard-deviation shift, and spectral mismatch, respectively. Larger values indicate stronger mismatch between the historical input and the future target.

| Horizon | ETTh1 | | | ETTh2 | | | ETTm1 | | | ETTm2 | | | Weather | | | Electricity | | | Traffic | | |
|---|---|---|---|---|---|---|---|---|---|---|---|---|---|---|---|---|---|---|---|---|---|
| | MS | SS | SM | MS | SS | SM | MS | SS | SM | MS | SS | SM | MS | SS | SM | MS | SS | SM | MS | SS | SM |
| $H = 96$ | 0.332 | 0.114 | 0.267 | 0.314 | 0.163 | 0.276 | 0.409 | 0.181 | 0.260 | 0.329 | 0.151 | 0.257 | 0.408 | 0.261 | 0.234 | 0.363 | 0.056 | 0.196 | 0.704 | 0.108 | 0.260 |
| $H = 192$ | 0.339 | 0.118 | 0.349 | 0.326 | 0.187 | 0.337 | 0.398 | 0.168 | 0.323 | 0.354 | 0.151 | 0.338 | 0.421 | 0.267 | 0.245 | 0.347 | 0.053 | 0.459 | 0.621 | 0.085 | 0.430 |
| $H = 336$ | 0.357 | 0.125 | 0.386 | 0.349 | 0.217 | 0.371 | 0.418 | 0.167 | 0.328 | 0.380 | 0.161 | 0.350 | 0.438 | 0.297 | 0.304 | 0.361 | 0.056 | 0.580 | 0.603 | 0.081 | 0.532 |
| $H = 720$ | 0.379 | 0.144 | 0.389 | 0.405 | 0.255 | 0.371 | 0.471 | 0.186 | 0.359 | 0.434 | 0.192 | 0.364 | 0.452 | 0.348 | 0.351 | 0.410 | 0.066 | 0.564 | 0.645 | 0.086 | 0.510 |

Here, $\mu_{\mathbf{X}}$ and $\mu_{\mathbf{Y}}$ denote the means of the historical and future windows, $\sigma_{\mathbf{X}}$ and $\sigma_{\mathbf{Y}}$ denote their standard deviations, and $\tilde{\mathbf{A}}_{\mathbf{X}}$ and $\tilde{\mathbf{A}}_{\mathbf{Y}}$ denote the normalized nonnegative spectra. MS and SS characterize distributional shifts in the time domain, whereas SM measures changes in frequency composition.

Table 6 shows that the history–future mismatch is widespread across datasets and prediction horizons. Even at $H = 96$, all datasets exhibit non-negligible discrepancies, indicating that the historical window already provides an imperfect coordinate system for the future target. As the prediction horizon increases, the mismatch generally becomes more pronounced, especially in the spectral domain. For example, Electricity shows a sharp increase in SM from 0.196 at $H = 96$ to 0.580 at $H = 336$, while Traffic increases from 0.260 to 0.532 over the same horizons. Weather exhibits a consistent rise across MS, SS, and SM, suggesting simultaneous changes in mean, variance, and frequency composition. The ETT datasets also show persistent mismatch across horizons, confirming that this phenomenon is not restricted to high-dimensional benchmarks.

This empirical evidence directly supports the Phase Amnesia motivation. Models that rely only on historical statistics or direct pattern reuse must extrapolate from a context whose distributional and spectral properties already deviate from the future target. Therefore, the forecasting difficulty does not arise solely from limited model capacity, but also from the mismatch between the historical coordinate system and the future phase trajectory. These observations motivate an explicit generative mechanism for future phase evolution, rather than passive reuse of historical statistics or fixed periodic patterns.

### B.4. Hyperparameter Analysis.

**Robustness to Global Period $W$: Validation of Generative Evolution (Hypothesis II).** As visualized in Figure 5, PULSE demonstrates exceptional robustness to the global period setting $W$. Across datasets, performance remains remarkably stable even when $W$ deviates from the ground-truth periodicity (e.g., $W = 167$ vs. $W = 24$ for hourly data). This empirical stability provides strong evidence for **Hypothesis II**: unlike traditional methods that rely

on passive "historical copying" or rigid frequency matching, our **Generative Phase Router** actively synthesizes the dynamical evolution operator. It learns to map arbitrary phase coordinates to correct future trajectories, effectively compensating for misaligned structural priors. Crucially, a theoretical boundary is observed only at $W = 1$, where performance degrades noticeably (e.g., in ETTh1 and Solar). Setting $W = 1$ collapses the structural reference frame, reducing the Phase Anchor to a static bias. This forces the model to model high-energy structural shifts entirely through the residual stream, triggering the **Variance Masking** phenomenon (Hypothesis I) where subtle stochastic fluctuations are suppressed. Thus, providing a valid disentanglement basis ($W > 1$) is the only prerequisite for activating PULSE's generative capability.

**Task-Dependent Sensitivity to codebook Size $L$.** The sensitivity analysis of codebook size $L$ (Figure 6) uncovers a fundamental distinction between "Schedule-Driven" and "Physics-Driven" systems. For schedule-driven domains like Electricity, performance is largely invariant to $L$. These systems follow rigid human calendars, where explicit timestamp cues (captured by our TimeEncoder) provide sufficient phase information, rendering the learnable codebook supplementary. In contrast, physical systems like Solar and ETTh1 exhibit high sensitivity, often requiring specific codebook resolutions (e.g., the optimal valley around $L \in [24, 48]$ for Solar). In these domains, dynamics are governed by complex physical interactions rather than strict clock time. Consequently, the TimeEncoder alone is insufficient, and the model relies heavily on the codebook to store and retrieve complex **geometric phase prototypes** (e.g., specific irradiance waveforms). This confirms the codebook's critical role as a non-parametric structural memory for capturing physical laws when explicit temporal cues are inadequate.

**Impact of Router Patch Size $P$ on Evolution Learning.** Figure 7 reveals a consistent preference for larger patch sizes ($P \in [16, 24]$) across complex datasets like ETTh1 and Solar. This trend directly validates the design of our Phase Router, which treats patches as semantic tokens to learn the continuous evolution operator. When $P$ is too small (e.g., $P = 4$), the phase tokens become overly fragmented, capturing only transient noise rather than the meaningful **local**

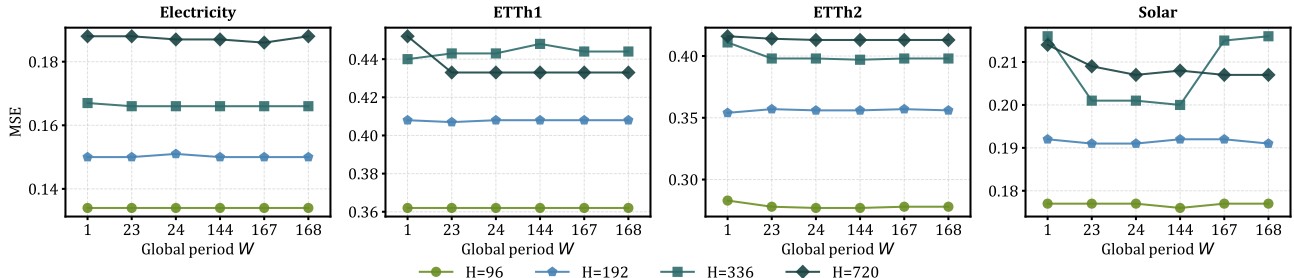

*Figure 5.* Impact of the global period $W$ on model performance across Electricity, ETTh1, ETTh2, and Solar.

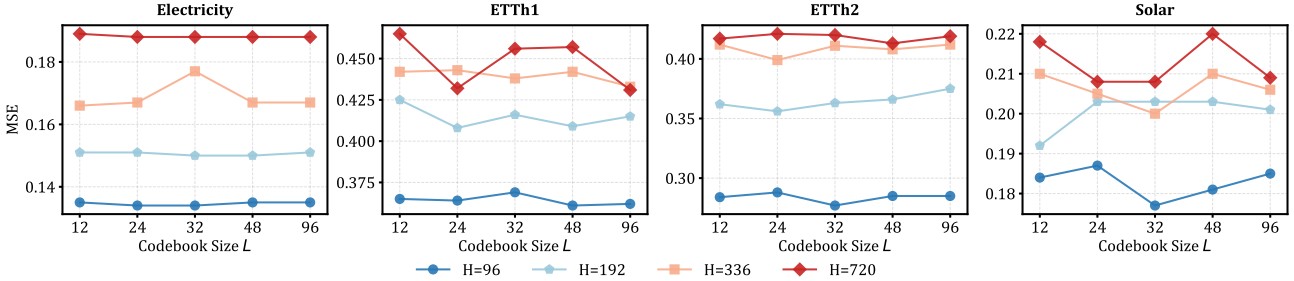

*Figure 6.* Sensitivity analysis regarding codebook size $L$ across Electricity, ETTh1, ETTh2, and Solar.

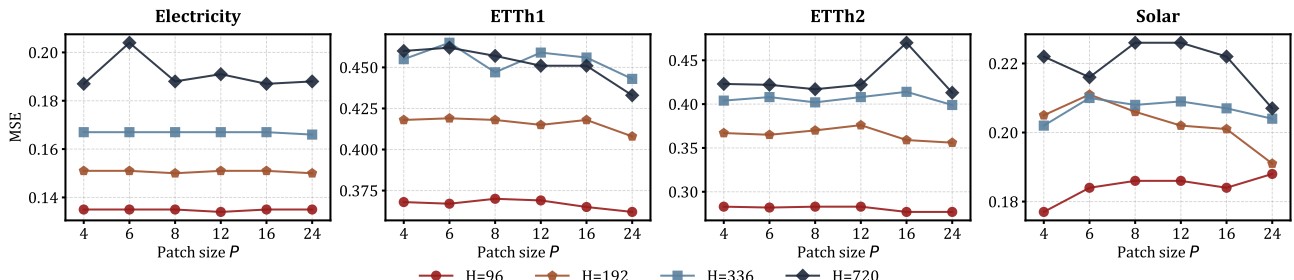

*Figure 7.* Performance evaluation under varying router patch sizes $P$ across Electricity, ETTh1, ETTh2, and Solar.

**trajectory geometry**. This creates a localized "Phase Amnesia," depriving the router of the context needed to extrapolate trends. Conversely, a larger patch size (e.g., $P = 24$) encapsulates a complete structural unit (e.g., a full diurnal cycle), enabling the router to operate on information-rich **Phase Anchors**. This allows for a more accurate estimation of the derivatives governing the dynamical system, thereby generating smoother and more physically consistent future trajectories.

### B.5. Distributional Priors in Statistic-Aware Mixup.

In the proposed Statistic-Aware Mixup (SAM), the interpolation coefficient $\lambda$ is sampled from a Beta distribution $\lambda \sim \mathrm{Beta}(\alpha, \alpha)$ (Zhang et al., 2018). The selection of this distribution and its hyperparameter $\alpha$ is grounded in the physical properties of time series residuals.

**Theoretical Motivation: The Manifold Constraint.** We employ the Beta distribution specifically for its topological flexibility within a bounded support. The critical design choice is the shape of the probability density. In time series forecasting, central mixing ($\lambda \approx 0.5$) often leads to *destructive interference*, where phase-mismatched signals cancel each other out, creating non-physical artifacts. To mitigate this, we enforce a **U-shaped prior** by setting $\alpha < 1$. This constraint concentrates probability mass near the boundaries, ensuring that the synthesized sample remains structurally dominated by one source signal (preserving the phase manifold (Verma et al., 2019)) while using the other merely as a statistical perturbation for regularization.

**Empirical Verification: Robustness and Degradation.** Sensitivity analysis across benchmarks (exemplified by ETTh2 in Figure 8) reveals a consistent phenomenon that validates our hypothesis:

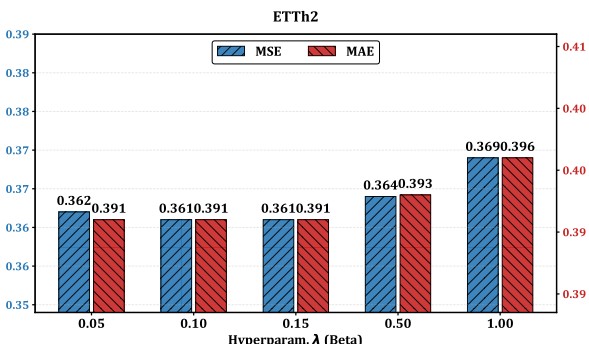

*Figure 8.* Sensitivity analysis of $\alpha$ on the ETTh2 dataset. The curve illustrates a general phenomenon: a stable **robustness plateau** in the U-shaped region ($\alpha \in [0.05, 0.15]$), followed by distinct degradation as the prior shifts towards a Uniform distribution ($\alpha = 1.0$).

- **The Robustness Plateau** ($\alpha \leq 0.25$): We observe a wide performance trough where the forecasting error remains minimal and invariant. Within the U-shaped region (e.g., $\alpha \in [0.05, 0.15]$), the model demonstrates high robustness, indicating that specific parameter tuning is unnecessary as long as the boundary-sampling property is maintained.

- **Degradation at Uniformity** ($\alpha \to 1.0$): A clear trend of performance degradation emerges as the distribution approaches uniformity ($\alpha = 1.0$). This empirical evidence confirms that frequent sampling of central weights corrupts the temporal structure of residuals.

Consequently, preserving phase integrity via a U-shaped prior is a prerequisite for effective non-stationary forecasting. Based on these findings, we set $\alpha = 0.15$ as the default, positioning it comfortably within the robust region.

### B.6. Spectral Analysis of Phase Anchor Evolution

In this section, we provide a deeper analysis of the qualitative differences observed in Figure 3 and Figure 10 regarding the behavior of the deterministic components: the Historical Phase Anchor ($\mathbf{A}_x$) and the Future Phase Anchor ($\mathbf{A}_y$).

**The Phenomenon: Intensity vs. Smoothness.** As illustrated in the time-domain visualizations of Figure 3 and Figure 10, a distinct morphological dichotomy exists between the historical and future anchors. The **High-Frequency Historical Anchoring** ($\mathbf{A}_x$), represented by the solid red and cyan lines in the upper panels, exhibits high-frequency oscillations and intense fluctuations. In many time steps, the amplitude of $\mathbf{A}_x$ matches or even momentarily exceeds the local variance of the raw input $X$. Conversely, the **Low-Frequency Future Evolution** ($\mathbf{A}_y$), shown in the lower

panels, manifests as a smooth, continuous trajectory. It effectively filters out high-frequency noise, retaining only the fundamental low-rank periodicity and trend modulation. This transformation from a "volatile" history to a "smooth" future is not an artifact, but a deliberate consequence of our Disentangle-Evolve-Simulate design philosophy. We analyze the theoretical necessity of this behavior below.

$\mathbf{A}_x$: **The Necessity of Energy Absorption.** The intense fluctuation of $\mathbf{A}_x$ is a prerequisite for satisfying Hypothesis I (Wold Decomposition (Wold, 1938)) and mitigating the Variance Masking problem formalized in Proposition 3.1. Its primary role is to function as a Structural Trap that absorbs the dominant, high-energy components of the signal. By aggressively modeling the instantaneous phase and local volatility, $\mathbf{A}_x$ ensures that the remaining residual $\mathbf{R}_x = X - \mathbf{A}_x$ is strictly stochastic and low-energy. Furthermore, this volatility prevents Optimization Interference (Koopmans, 1995). If $\mathbf{A}_x$ were overly smooth, significant structural trends would leak into the residual $\mathbf{R}_x$. This would inflate the variance $\sigma_\mathbf{R}$, causing the standard normalization (RevIN (Kim et al., 2021)) to suppress the gradients of subtle fluctuations (scaling by $O(\sigma_{\text{trend}}^{-1})$ instead of $O(\sigma_{\text{noise}}^{-1})$). Therefore, $\mathbf{A}_x$ must be volatile to fully capture the "gross" dynamics, leaving the backbone to focus purely on the fine-grained residual manifold.

$\mathbf{A}_y$: **The Manifestation of Deterministic Laws.** The transition to a smooth $\mathbf{A}_y$ validates Hypothesis II (Dynamical Phase Evolution). This smoothness signifies the recovery of the **intrinsic low-rank structure** from noisy observations. According to the **Low-Rank Hypothesis** (Huang et al., 2025), while the raw historical data $X$ exhibits high-rank characteristics due to stochastic noise corruption, deterministic phase evolution is inherently dominated by sparse basis dependencies. Consequently, the Phase Router performs a critical process of information condensation: by projecting the continuous time series into a compact phase resolution, it filters out high-rank noise and synthesizes a future trajectory $\mathbf{A}_y$ that aligns with the structural low-rank subspace. From the frequency-domain perspective, this condensation is reflected in Figure 9, where $\mathbf{A}_y$ exhibits a substantially lower-amplitude spectrum than the future target $\mathbf{Y}$. This indicates that $\mathbf{A}_y$ is not intended to reconstruct the full spectral energy of $\mathbf{Y}$; instead, it extracts a compact deterministic spectral backbone, while the transient and stochastic variations are preserved in $\mathbf{R}_y$. In this sense, the attention mechanism acts as a learnable low-pass filter, ensuring that the generated anchor represents the expected evolution path rather than stochastic fluctuations. The smoothness of $\mathbf{A}_y$ therefore indicates that the model has successfully disentangled the evolving low-rank trend from the remaining high-rank residual variations.

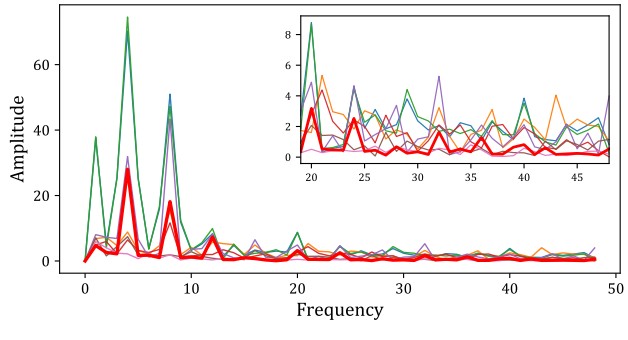
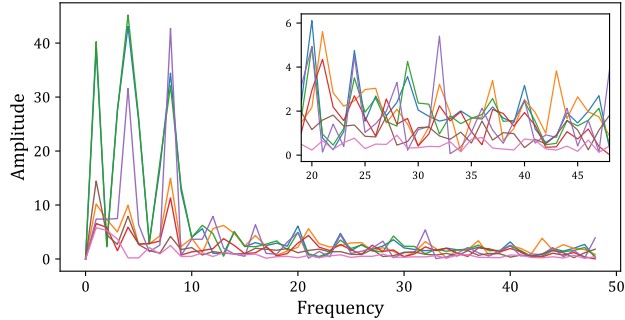

(a) Future time series $Y$ spectrum (colorful lines)
and deterministic part $A_y$ spectrum (bold red line)

(b) Future residual stochastic part $R_y$ spectrum

*Figure 9.* **Frequency-domain visualization of future-side disentanglement.** Panel (a) compares the spectra of the future target $\mathbf{Y}$ and the deterministic future anchor $\mathbf{A}_y$, while Panel (b) shows the spectrum of the stochastic residual $\mathbf{R}_y$. $\mathbf{A}_y$ extracts a low-amplitude deterministic spectral backbone, whereas $\mathbf{R}_y$ captures the remaining stochastic variations.

*Table 7.* **Effect of scaling up** $d_{\text{model}}$. Each metric is averaged over prediction horizons $H \in \{96, 192, 336, 720\}$. The compact default setting $d_{\text{model}} = 16/32$ achieves the best overall performance.

| $d_{\text{model}}$ | ETTh1 | | ETTh2 | | ETTm1 | | ETTm2 | | Electricity | | Weather | |
|---|---|---|---|---|---|---|---|---|---|---|---|---|
| | MSE | MAE | MSE | MAE | MSE | MAE | MSE | MAE | MSE | MAE | MSE | MAE |
| 16/32 | **0.412** | **0.422** | **0.361** | **0.391** | **0.363** | **0.388** | **0.262** | **0.314** | **0.160** | **0.255** | **0.239** | **0.271** |
| 64 | 0.420 | 0.425 | 0.436 | 0.395 | 0.368 | 0.390 | 0.268 | 0.317 | 0.164 | 0.257 | 0.242 | 0.274 |
| 128 | 0.421 | 0.426 | 0.370 | 0.397 | 0.368 | 0.391 | 0.268 | 0.317 | 0.164 | 0.257 | 0.243 | 0.276 |
| 256 | 0.426 | 0.426 | 0.377 | 0.401 | 0.401 | 0.404 | 0.279 | 0.325 | 0.185 | 0.279 | 0.274 | 0.304 |
| 512 | 0.593 | 0.509 | 0.416 | 0.432 | 0.435 | 0.425 | 0.297 | 0.334 | 0.203 | 0.298 | 0.279 | 0.309 |

## B.7. Computational Complexity and Efficiency Analysis.

**Length-Invariant Attention via Phase Tokenization.** Conventional Transformers suffer from token proliferation, where a sequence of length $T$ yields $N \propto T$ tokens, inevitably leading to quadratic attention costs $\mathbf{O}(T^2)$ (Vaswani et al., 2017). Even patch-based methods (Nie et al., 2023) remain bound by linear token growth. PULSE fundamentally resolves this bottleneck via **Phase Tokenization**. Instead of slicing time, we project the series into a fixed phase space defined by a tunable resolution $P$ (where $P \in \{4, \ldots, 24\}$). This strategy enforces the token count to be a structural constant, independent of the look-back window $T$. Consequently, the Phase Router operates on a fixed $P \times P$ interaction matrix. Since $P \ll T$, the complexity of the attention mechanism is reduced to $\mathbf{O}(P^2)$, which effectively behaves as $\mathbf{O}(1)$ relative to the input length. This creates a "zero-latency" global mixer that completely decouples the computational cost of modeling dependencies from the sequence length.

**Efficiency Origins: The Low-Rank Hypothesis.** The computational efficiency of PULSE is not an engineered coincidence but a direct derivation from the **Low-Rank Hypothesis** (Huang et al., 2025). We posit that deterministic phase evolution resides on a highly compact manifold dominated by sparse basis dependencies. Crucially, our Phase

Tokenization captures this structure by learning **condensed phase relationships**. By projecting the continuous time series into a compact phase resolution $P$, the model is forced to distill scattered and redundant temporal observations into a few dense phase tokens. This information condensation filters out high-rank noise and aligns the representation space with the intrinsic low-rank manifold. The capacity-scaling results in Table 7 further support this assumption: increasing the auxiliary hidden dimension $d_{\text{model}}$ beyond the compact default setting does not yield systematic gains, but instead often leads to degraded performance. This suggests that the dominant phase-evolution dynamics are already sufficiently represented in a low-dimensional latent space, while excessive channel capacity introduces redundant degrees of freedom that may disturb the compact phase structure.

This theoretical alignment fundamentally obviates the need for over-parameterized capacity, enabling a **synergistic reduction in both width and depth**. Specifically, since the tokens encode condensed phase dependencies with high information density, extremely narrow channels ($d_{\text{model}} \in \{16, 32\}$) are sufficient to encode the evolution, reducing element-wise FLOPs by orders of magnitude compared to standard forecasters with $d_{\text{model}} = 512$ (Nie et al., 2023; Liu et al., 2024; Dai et al., 2024a). **Simultaneously**, this condensed global view renders deep hierarchical abstraction unnecessary. Unlike high-rank patterns that require deep stacking, typically 2–4 layers,

*Table 8.* **Extended look-back forecasting results.** $T$ denotes the look-back window length, and each metric is averaged over $H \in \{96, 192, 336, 720\}$. Lower values indicate better performance.

| $T$ | Dataset Model | ETTh1 MSE | ETTh1 MAE | ETTh2 MSE | ETTh2 MAE | ETTm1 MSE | ETTm1 MAE | ETTm2 MSE | ETTm2 MAE | Electricity MSE | Electricity MAE | Solar MSE | Solar MAE | Traffic MSE | Traffic MAE | Weather MSE | Weather MAE |
|---|---|---|---|---|---|---|---|---|---|---|---|---|---|---|---|---|---|
| | PULSE | **0.409** | **0.425** | **0.339** | **0.384** | **0.344** | **0.374** | **0.249** | **0.307** | **0.149** | **0.246** | 0.199 | **0.236** | **0.409** | 0.276 | **0.222** | **0.262** |
| 336 | CFPT | 0.423 | 0.435 | 0.341 | 0.385 | 0.348 | 0.376 | 0.260 | 0.317 | 0.155 | 0.249 | 0.207 | 0.256 | **0.409** | 0.277 | 0.225 | **0.262** |
| | CycleNet | 0.415 | 0.426 | 0.355 | 0.398 | 0.355 | 0.379 | 0.251 | 0.309 | 0.158 | 0.250 | **0.194** | 0.255 | 0.413 | 0.281 | 0.226 | 0.266 |
| | PULSE | **0.410** | **0.427** | **0.335** | **0.385** | **0.342** | **0.374** | **0.248** | **0.308** | **0.150** | 0.249 | **0.194** | **0.234** | 0.406 | 0.277 | **0.220** | 0.260 |
| 512 | CFPT | 0.416 | 0.432 | 0.339 | 0.388 | 0.352 | 0.379 | 0.262 | 0.319 | 0.155 | **0.249** | 0.197 | 0.252 | **0.401** | **0.272** | 0.223 | 0.262 |
| | CycleNet | 0.423 | 0.434 | 0.347 | 0.394 | 0.358 | 0.382 | 0.252 | 0.311 | 0.157 | 0.251 | 0.211 | 0.267 | 0.405 | 0.278 | 0.227 | **0.257** |
| | PULSE | **0.409** | **0.428** | **0.342** | **0.388** | **0.342** | **0.377** | **0.247** | **0.308** | **0.149** | 0.250 | 0.191 | **0.235** | 0.398 | 0.275 | **0.219** | **0.261** |
| 720 | CFPT | 0.426 | 0.441 | 0.347 | 0.395 | 0.353 | 0.381 | 0.263 | 0.322 | 0.155 | **0.250** | 0.212 | 0.260 | **0.398** | **0.274** | 0.243 | 0.276 |
| | CycleNet | 0.430 | 0.439 | 0.345 | 0.394 | 0.355 | 0.381 | 0.249 | 0.312 | 0.157 | **0.250** | **0.189** | 0.247 | 0.403 | 0.282 | 0.224 | 0.266 |

*Table 9.* **Plug-and-play results on the Electricity dataset with $T = 720$.** $T$ denotes the look-back window length and $H$ denotes the prediction horizon. Lower MSE and MAE indicate better forecasting performance.

| $H$ | iTransformer Original MSE | iTransformer Original MAE | iTransformer +PULSE MSE | iTransformer +PULSE MAE | DLinear Original MSE | DLinear Original MAE | DLinear +PULSE MSE | DLinear +PULSE MAE | TimesNet Original MSE | TimesNet Original MAE | TimesNet +PULSE MSE | TimesNet +PULSE MAE | SparseTSF Original MSE | SparseTSF Original MAE | SparseTSF +PULSE MSE | SparseTSF +PULSE MAE | PatchTST Original MSE | PatchTST Original MAE | PatchTST +PULSE MSE | PatchTST +PULSE MAE |
|---|---|---|---|---|---|---|---|---|---|---|---|---|---|---|---|---|---|---|---|---|
| 96 | 0.135 | 0.233 | **0.127** | **0.224** | 0.141 | 0.244 | **0.129** | **0.224** | 0.202 | 0.308 | **0.178** | **0.284** | 0.139 | 0.239 | **0.132** | **0.223** | 0.141 | 0.240 | **0.130** | **0.227** |
| 192 | 0.155 | 0.253 | **0.147** | **0.250** | 0.155 | 0.258 | **0.150** | **0.245** | 0.218 | 0.322 | **0.193** | **0.299** | 0.155 | 0.250 | **0.147** | **0.240** | 0.156 | 0.256 | **0.146** | **0.243** |
| 336 | 0.169 | 0.267 | **0.161** | **0.266** | 0.170 | 0.275 | **0.165** | **0.268** | 0.232 | 0.332 | **0.208** | **0.320** | 0.171 | 0.265 | **0.158** | **0.259** | 0.172 | 0.267 | **0.163** | **0.261** |
| 720 | 0.204 | 0.301 | **0.173** | **0.280** | 0.209 | 0.309 | **0.181** | **0.285** | 0.299 | 0.375 | **0.240** | **0.344** | 0.208 | 0.300 | **0.184** | **0.285** | 0.208 | 0.299 | **0.184** | **0.294** |

to gradually expand receptive fields, our low-rank phase tokens inherently provide a global structure. Consequently, a **single layer** of interaction in the Phase Router is sufficient to capture the dominant evolution dynamics, avoiding the latency and memory-access overhead of deep networks.

**Overall Complexity Formulation.** The total computational complexity is formally derived as the sum of the linear projection layers and the constant-cost router attention:

$$\Omega(\text{PULSE}) = \underbrace{\mathbf{O}(T)}_{\text{Linear Projections}} + \underbrace{\mathbf{O}(P^2)}_{\text{Phase Router}} \quad (11)$$

Given that the phase resolution $P$ is a small constant (max 24) and $d_{model}$ is minimized ($16 \sim 32$) due to the low-rank constraint, the quadratic attention term becomes mathematically negligible compared to the linear term. The system is thus dominated by the lightweight linear projections, resulting in an asymptotically linear complexity $\mathbf{O}(T)$. This theoretically confirms that PULSE achieves the inference speed of linear models while retaining the non-linear generative capabilities of the attention mechanism within the low-rank phase subspace.

## C. More Results of PULSE

### C.1. Extended Forecasting Analysis

Table 12 presents the comprehensive performance of PULSE against state-of-the-art baselines under the standard look-back setting $T = 96$. Beyond aggregate metrics, a deeper inspection of the error degradation trend reveals the root cause of PULSE's superiority: its resistance to error accumulation over long prediction horizons. While conventional models often suffer from rapid performance decay, or "Scale Collapse", as the forecasting horizon $H$ extends to 720, PULSE maintains a significantly flatter degradation trajectory. A representative example is the volatile **Solar** dataset, where runner-up baselines exhibit pronounced error increases at long horizons, whereas PULSE preserves more stable predictive accuracy. This suggests that the proposed Phase-Anchored Disentanglement does not merely fit short-term local patterns, but actively generates a physically consistent future manifold, thereby mitigating the variance collapse commonly observed in extended forecasting.

To further examine whether this advantage depends on the standard short-context setting, we additionally evaluate longer look-back windows. The look-back length of 96 follows the standard benchmark protocol used in prior studies, while Table 8 reports the results under longer historical contexts, namely $T \in \{336, 512, 720\}$. For each dataset, the three row groups correspond to look-back lengths 336,

*Table 10.* **Additional forecasting results with MSE, MAE, and MASE.** Results are reported on five normal-scale datasets and one large-scale dataset with $T = 96$. Each metric is averaged over prediction horizons $H \in \{96, 192, 336, 720\}$, and MASE is computed with seasonal period $m = 1$. Lower values indicate better performance.

| Dataset | SD | | | AQShunyi | | | AQWan | | | ZafNoo | | | CzeLan | | | Wind | | |
|---|---|---|---|---|---|---|---|---|---|---|---|---|---|---|---|---|---|---|
| Model | MSE | MAE | MASE | MSE | MAE | MASE | MSE | MAE | MASE | MSE | MAE | MASE | MSE | MAE | MASE | MSE | MAE | MASE |
| PULSE | 1.299 | 0.201 | 1.597 | 0.734 | 0.517 | 3.469 | 0.841 | 0.510 | 3.516 | 0.564 | 0.461 | 2.558 | 0.242 | 0.282 | 4.576 | 0.977 | 0.688 | 6.457 |
| CFPT | 1.493 | 0.209 | 1.663 | 0.779 | 0.526 | 3.527 | 0.887 | 0.516 | 3.554 | 0.566 | 0.462 | 2.563 | 0.257 | 0.290 | 4.871 | 1.018 | 0.717 | 6.728 |
| CycleNet | 1.341 | 0.254 | 2.027 | 0.764 | 0.521 | 3.494 | 0.868 | 0.512 | 3.526 | 0.567 | 0.462 | 2.563 | 0.251 | 0.284 | 4.605 | 0.998 | 0.706 | 6.628 |

*Table 11.* **Plug-and-play MASE results with $T = 720$.** $T$ denotes the look-back window length. The columns 96, 192, 336, and 720 denote prediction horizons. MASE is computed with seasonal period $m = 96$. Best results between each backbone and its +PULSE variant are highlighted in **bold**.

| Dataset | Wind | | | | AQShunyi | | | | AQWan | | | |
|---|---|---|---|---|---|---|---|---|---|---|---|---|
| Model | 96 | 192 | 336 | 720 | 96 | 192 | 336 | 720 | 96 | 192 | 336 | 720 |
| iTransformer | 0.747 | 0.836 | 0.896 | 0.931 | 0.760 | 0.797 | 0.817 | 0.875 | 0.780 | 0.806 | 0.833 | 0.888 |
| +PULSE | **0.689** | **0.788** | **0.846** | **0.900** | **0.731** | **0.770** | **0.781** | **0.813** | **0.736** | **0.764** | **0.821** | **0.832** |
| DLinear | 0.750 | 0.809 | 0.861 | **0.894** | 0.752 | 0.789 | 0.820 | 0.911 | 0.769 | 0.809 | 0.833 | 0.901 |
| +PULSE | **0.716** | **0.778** | **0.842** | 0.895 | **0.731** | **0.768** | **0.797** | **0.863** | **0.730** | **0.768** | **0.817** | **0.875** |
| TimesNet | 0.793 | 0.914 | 0.982 | 1.062 | 0.755 | 0.790 | 0.802 | **0.842** | 0.839 | 0.861 | 0.824 | **0.822** |
| +PULSE | **0.729** | **0.802** | **0.849** | **0.919** | **0.734** | **0.787** | **0.799** | 0.849 | **0.736** | **0.792** | **0.803** | 0.850 |
| PatchTST | 0.760 | 0.848 | 0.882 | 0.924 | 0.762 | 0.802 | 0.821 | 0.855 | 0.769 | 0.805 | 0.819 | 0.893 |
| +PULSE | **0.691** | **0.795** | **0.860** | **0.904** | **0.741** | **0.782** | **0.799** | **0.852** | **0.743** | **0.776** | **0.802** | **0.862** |

512, and 720, respectively, and each metric is averaged over prediction horizons $H \in \{96, 192, 336, 720\}$. The results show that PULSE remains effective beyond the short-context setting and continues to achieve strong overall performance across diverse datasets. Notably, longer histories do not invalidate the advantage of PULSE; instead, the model consistently benefits from its phase-evolution mechanism, which extracts stable deterministic structure while suppressing redundant or noisy historical variations. This further supports our central claim that PULSE improves long-term forecasting by modeling future phase evolution, rather than by relying on direct historical copying or increasing the look-back window alone.

## C.2. Deep Dive into Plug-and-Play Dynamics

This intrinsic capability to mitigate long-term error accumulation is not limited to PULSE as a standalone model; it also transfers effectively to other architectures when deployed as a plug-and-play module. As detailed in Table 13, this effect is most evident in challenging long-term forecasting scenarios where backbone models typically suffer from substantial error accumulation.

To further isolate this behavior, Table 9 reports plug-and-play results on the Electricity dataset with a long look-back window $T = 720$. The results consistently show that adding PULSE improves different architectural families, including Transformer-based, linear, convolutional, and sparse fore-

casting models. More importantly, the improvement persists across all prediction horizons, indicating that PULSE does not merely provide a short-horizon correction but acts as a stable phase-aware refinement mechanism. As the horizon extends to $H = 720$, the gains become more pronounced for several backbones, suggesting that PULSE effectively suppresses long-term error accumulation by decoupling stable periodic laws from evolving residual variations. Therefore, PULSE functions as a general plug-and-play statistical correction layer, enhancing the robustness of existing forecasters under extended forecasting windows.

## C.3. Additional MASE-based Evaluation

Beyond MSE and MAE, we further report MASE to evaluate whether the improvement of PULSE remains consistent under a scale-normalized metric. MASE normalizes the forecasting error by the error of a seasonal naive baseline:

$$\text{MASE} = \frac{\frac{1}{n}\sum_{t=1}^{n}|y_t - \hat{y}_t|}{\frac{1}{n-m}\sum_{t=m+1}^{n}|y_t - y_{t-m}| + \epsilon}, \quad (12)$$

where $m$ is the seasonal period and $\epsilon$ is a small constant for numerical stability.

We consider two MASE settings. In Table 10, MASE is computed with $m = 1$ under the standard look-back window $T = 96$, and all metrics are averaged over prediction horizons $H \in \{96, 192, 336, 720\}$. In Table 11, MASE is computed with $m = 96$ under the longer look-back window

$T = 720$ to evaluate plug-and-play transferability. When $m = 1$, MASE values greater than one are expected and should not be interpreted as an isolated failure. This is because all models perform direct multi-step forecasting in a one-shot manner, while the naive reference with $m = 1$ relies on immediate lag-one observations and can therefore be relatively strong. Thus, the key comparison is whether different methods are evaluated under the same protocol, where lower MASE indicates better normalized forecasting accuracy.

Table 10 reports results on five normal-scale datasets from TFB, including AQShunyi, AQWan, ZafNoo, CzeLan, and Wind (Qiu et al., 2024), together with the large-scale SD dataset from LargeST (Liu et al., 2023a). Across all six datasets, PULSE consistently achieves the best MSE, MAE, and MASE. This shows that the advantage of PULSE is not limited to scale-dependent metrics, but also holds when the error is normalized by a seasonal naive baseline. The gains are especially meaningful on challenging datasets such as CzeLan and Wind, where relatively large MASE values indicate a stronger naive reference and a more demanding normalized forecasting task.

Table 11 further evaluates whether this normalized improvement transfers to existing backbone models. On Wind, AQShunyi, and AQWan with $T = 720$, adding PULSE reduces MASE across nearly all backbone–horizon combinations. This confirms that PULSE not only performs strongly as a standalone forecaster, but also serves as a general plug-and-play correction module that improves normalized forecasting accuracy for diverse backbones.

## D. More Visualization Results

In this section, we provide extended qualitative visualizations to demonstrate the effectiveness of PULSE. Specifically, Figure 10 presents the T-SNE analysis of the learned phase anchors compared with raw data. Figure 11 provides additional forecasting visualizations on Electricity, Solar, Traffic, and Weather. Figure 12 provides a comprehensive comparison of the plug-and-play forecasting results across four baseline models (DLinear, PatchTST, TimesNet, and iTransformer) on the Electricity dataset.

*Table 12.* Full multivariate time series forecasting results of Table 1 across all four prediction horizons ($H \in \{96, 192, 336, 720\}$). The look-back length is fixed at 96. We reproduce the results for CFPT (Kou et al., 2025), TimeXer (Wang et al., 2024b), and CycleNet (Lin et al., 2024), while other baseline results are sourced from iTransformer (Liu et al., 2024). The best results are highlighted in **bold**, and the second-best are underlined.

| Model | | PULSE (ours) | | CFPT (2025) | | TimeXer (2024b) | | CycleNet (2024) | | iTransformer (2024) | | MSGNet (2024) | | TimesNet (2023) | | PatchTST (2023) | | Crossformer (2023) | | DLinear (2023) | |
|---|---|---|---|---|---|---|---|---|---|---|---|---|---|---|---|---|---|---|---|---|---|
| Metric | | MSE | MAE | MSE | MAE | MSE | MAE | MSE | MAE | MSE | MAE | MSE | MAE | MSE | MAE | MSE | MAE | MSE | MAE | MSE | MAE |
| ETTh1 | 96 | **0.362** | **0.390** | 0.372 | 0.391 | 0.382 | 0.403 | 0.375 | 0.395 | 0.386 | 0.405 | 0.390 | 0.411 | 0.384 | 0.402 | 0.414 | 0.419 | 0.423 | 0.448 | 0.386 | 0.400 |
| | 192 | **0.408** | **0.418** | 0.425 | 0.421 | 0.429 | 0.435 | 0.436 | 0.428 | 0.441 | 0.436 | 0.443 | 0.442 | 0.436 | 0.429 | 0.460 | 0.445 | 0.471 | 0.474 | 0.437 | 0.432 |
| | 336 | **0.443** | **0.433** | 0.467 | 0.442 | 0.468 | 0.448 | 0.496 | 0.455 | 0.487 | 0.458 | 0.482 | 0.469 | 0.491 | 0.469 | 0.501 | 0.466 | 0.570 | 0.546 | 0.481 | 0.459 |
| | 720 | **0.433** | **0.448** | 0.466 | 0.461 | 0.469 | 0.461 | 0.520 | 0.484 | 0.503 | 0.491 | 0.496 | 0.488 | 0.521 | 0.500 | 0.500 | 0.488 | 0.653 | 0.621 | 0.519 | 0.516 |
| | Avg | **0.412** | **0.422** | 0.433 | 0.429 | 0.437 | 0.437 | 0.457 | 0.441 | 0.454 | 0.447 | 0.453 | 0.453 | 0.458 | 0.450 | 0.469 | 0.455 | 0.529 | 0.522 | 0.456 | 0.452 |
| ETTh2 | 96 | **0.277** | **0.330** | 0.285 | 0.336 | 0.286 | 0.338 | 0.298 | 0.344 | 0.297 | 0.349 | 0.329 | 0.371 | 0.340 | 0.374 | 0.302 | 0.348 | 0.745 | 0.584 | 0.333 | 0.387 |
| | 192 | **0.356** | **0.381** | 0.363 | 0.388 | 0.363 | 0.389 | 0.372 | 0.396 | 0.380 | 0.400 | 0.402 | 0.414 | 0.402 | 0.414 | 0.388 | 0.400 | 0.877 | 0.656 | 0.477 | 0.476 |
| | 336 | **0.399** | **0.417** | 0.413 | 0.426 | 0.414 | 0.423 | 0.431 | 0.439 | 0.428 | 0.432 | 0.440 | 0.445 | 0.452 | 0.452 | 0.426 | 0.433 | 1.043 | 0.731 | 0.594 | 0.541 |
| | 720 | 0.413 | 0.436 | **0.396** | **0.422** | 0.408 | 0.432 | 0.450 | 0.458 | 0.427 | 0.445 | 0.480 | 0.477 | 0.462 | 0.468 | 0.431 | 0.446 | 1.104 | 0.763 | 0.831 | 0.657 |
| | Avg | **0.361** | **0.391** | 0.364 | 0.393 | 0.368 | 0.396 | 0.388 | 0.409 | 0.383 | 0.407 | 0.413 | 0.427 | 0.414 | 0.427 | 0.387 | 0.407 | 0.942 | 0.684 | 0.559 | 0.515 |
| ETTm1 | 96 | **0.303** | **0.347** | 0.316 | 0.356 | 0.318 | 0.356 | 0.319 | 0.360 | 0.334 | 0.368 | 0.319 | 0.366 | 0.338 | 0.375 | 0.329 | 0.367 | 0.404 | 0.426 | 0.345 | 0.372 |
| | 192 | **0.349** | **0.379** | 0.354 | 0.380 | 0.362 | 0.383 | 0.360 | 0.381 | 0.377 | 0.391 | 0.377 | 0.397 | 0.374 | 0.387 | 0.367 | 0.385 | 0.450 | 0.451 | 0.380 | 0.389 |
| | 336 | **0.375** | **0.398** | 0.383 | 0.400 | 0.395 | 0.407 | 0.389 | 0.403 | 0.426 | 0.420 | 0.417 | 0.422 | 0.410 | 0.411 | 0.399 | 0.410 | 0.532 | 0.515 | 0.413 | 0.413 |
| | 720 | **0.424** | **0.426** | 0.444 | 0.434 | 0.452 | 0.441 | 0.447 | 0.441 | 0.491 | 0.459 | 0.487 | 0.463 | 0.478 | 0.450 | 0.454 | 0.439 | 0.666 | 0.589 | 0.474 | 0.453 |
| | Avg | **0.363** | **0.388** | 0.374 | 0.393 | 0.382 | 0.397 | 0.379 | 0.396 | 0.407 | 0.410 | 0.400 | 0.412 | 0.400 | 0.406 | 0.387 | 0.400 | 0.513 | 0.495 | 0.403 | 0.407 |
| ETTm2 | 96 | **0.163** | 0.248 | 0.167 | 0.249 | 0.171 | 0.256 | **0.163** | **0.246** | 0.180 | 0.264 | 0.182 | 0.266 | 0.187 | 0.267 | 0.175 | 0.259 | 0.287 | 0.366 | 0.193 | 0.292 |
| | 192 | **0.227** | 0.291 | 0.232 | 0.292 | 0.237 | 0.299 | 0.229 | **0.290** | 0.250 | 0.309 | 0.248 | 0.306 | 0.249 | 0.309 | 0.241 | 0.302 | 0.414 | 0.492 | 0.284 | 0.362 |
| | 336 | **0.283** | 0.330 | 0.290 | 0.331 | 0.296 | 0.338 | 0.284 | **0.327** | 0.311 | 0.348 | 0.312 | 0.346 | 0.321 | 0.351 | 0.305 | 0.343 | 0.597 | 0.542 | 0.369 | 0.427 |
| | 720 | **0.374** | **0.387** | 0.385 | 0.389 | 0.392 | 0.394 | 0.389 | 0.391 | 0.412 | 0.407 | 0.414 | 0.404 | 0.408 | 0.403 | 0.402 | 0.400 | 1.730 | 1.042 | 0.554 | 0.522 |
| | Avg | **0.262** | **0.314** | 0.269 | 0.315 | 0.274 | 0.322 | 0.266 | **0.314** | 0.288 | 0.332 | 0.289 | 0.330 | 0.291 | 0.333 | 0.281 | 0.326 | 0.757 | 0.611 | 0.350 | 0.401 |
| Electricity | 96 | **0.134** | **0.228** | 0.136 | 0.231 | 0.140 | 0.242 | 0.136 | 0.229 | 0.148 | 0.240 | 0.165 | 0.274 | 0.168 | 0.272 | 0.181 | 0.270 | 0.219 | 0.314 | 0.197 | 0.282 |
| | 192 | **0.150** | **0.243** | 0.153 | 0.246 | 0.157 | 0.256 | 0.152 | 0.244 | 0.162 | 0.253 | 0.185 | 0.292 | 0.184 | 0.289 | 0.188 | 0.274 | 0.231 | 0.322 | 0.196 | 0.285 |
| | 336 | **0.166** | **0.261** | 0.168 | 0.265 | 0.176 | 0.275 | 0.170 | 0.264 | 0.178 | 0.269 | 0.197 | 0.304 | 0.198 | 0.300 | 0.204 | 0.293 | 0.246 | 0.337 | 0.209 | 0.301 |
| | 720 | **0.188** | **0.289** | 0.199 | 0.293 | 0.211 | 0.306 | 0.212 | 0.299 | 0.225 | 0.317 | 0.231 | 0.332 | 0.220 | 0.320 | 0.246 | 0.324 | 0.280 | 0.363 | 0.245 | 0.333 |
| | Avg | **0.160** | **0.255** | 0.164 | 0.259 | 0.171 | 0.270 | 0.168 | 0.259 | 0.178 | 0.270 | 0.194 | 0.301 | 0.193 | 0.295 | 0.205 | 0.290 | 0.244 | 0.334 | 0.212 | 0.300 |
| Solar | 96 | **0.176** | **0.229** | 0.197 | 0.242 | 0.215 | 0.295 | 0.190 | 0.247 | 0.203 | 0.237 | 0.210 | 0.246 | 0.250 | 0.292 | 0.234 | 0.286 | 0.310 | 0.331 | 0.290 | 0.378 |
| | 192 | **0.192** | **0.245** | 0.227 | 0.261 | 0.236 | 0.301 | 0.210 | 0.266 | 0.233 | 0.261 | 0.265 | 0.290 | 0.296 | 0.318 | 0.267 | 0.310 | 0.734 | 0.725 | 0.320 | 0.398 |
| | 336 | **0.200** | **0.247** | 0.248 | 0.277 | 0.252 | 0.307 | 0.217 | 0.266 | 0.248 | 0.273 | 0.294 | 0.318 | 0.319 | 0.330 | 0.290 | 0.315 | 0.750 | 0.735 | 0.353 | 0.415 |
| | 720 | **0.208** | **0.249** | 0.259 | 0.282 | 0.244 | 0.305 | 0.223 | 0.266 | 0.249 | 0.275 | 0.285 | 0.315 | 0.338 | 0.337 | 0.289 | 0.317 | 0.769 | 0.765 | 0.356 | 0.413 |
| | Avg | **0.194** | **0.243** | 0.233 | 0.267 | 0.237 | 0.302 | 0.210 | 0.261 | 0.233 | 0.262 | 0.263 | 0.292 | 0.301 | 0.319 | 0.270 | 0.307 | 0.641 | 0.639 | 0.330 | 0.401 |
| Traffic | 96 | 0.414 | **0.268** | 0.444 | 0.274 | 0.428 | 0.271 | 0.458 | 0.296 | **0.395** | 0.268 | 0.608 | 0.349 | 0.593 | 0.321 | 0.462 | 0.290 | 0.522 | 0.290 | 0.650 | 0.396 |
| | 192 | 0.444 | 0.278 | 0.460 | 0.280 | 0.448 | 0.282 | 0.457 | 0.294 | **0.417** | **0.276** | 0.634 | 0.371 | 0.617 | 0.336 | 0.466 | 0.290 | 0.530 | 0.293 | 0.598 | 0.370 |
| | 336 | 0.458 | 0.287 | 0.477 | 0.289 | 0.473 | 0.289 | 0.470 | 0.299 | **0.433** | **0.283** | 0.669 | 0.388 | 0.629 | 0.336 | 0.482 | 0.300 | 0.558 | 0.305 | 0.605 | 0.373 |
| | 720 | 0.525 | 0.312 | 0.499 | 0.313 | 0.516 | 0.307 | 0.502 | 0.314 | **0.467** | **0.302** | 0.729 | 0.420 | 0.640 | 0.350 | 0.514 | 0.320 | 0.589 | 0.328 | 0.645 | 0.394 |
| | Avg | 0.460 | 0.286 | 0.470 | 0.289 | 0.466 | 0.287 | 0.472 | 0.301 | **0.428** | **0.282** | 0.660 | 0.382 | 0.620 | 0.336 | 0.481 | 0.300 | 0.550 | 0.304 | 0.625 | 0.383 |
| Weather | 96 | **0.150** | **0.196** | 0.154 | 0.200 | 0.157 | 0.205 | 0.158 | 0.203 | 0.174 | 0.214 | 0.163 | 0.212 | 0.172 | 0.220 | 0.177 | 0.210 | 0.158 | 0.230 | 0.196 | 0.255 |
| | 192 | **0.200** | 0.245 | 0.203 | **0.242** | 0.204 | 0.247 | 0.207 | 0.247 | 0.221 | 0.254 | 0.211 | 0.254 | 0.219 | 0.261 | 0.225 | 0.250 | 0.206 | 0.277 | 0.237 | 0.296 |
| | 336 | **0.260** | 0.291 | 0.261 | **0.286** | 0.261 | 0.290 | 0.262 | 0.289 | 0.278 | 0.296 | 0.273 | 0.299 | 0.280 | 0.306 | 0.278 | 0.290 | 0.272 | 0.335 | 0.283 | 0.335 |
| | 720 | 0.347 | 0.353 | 0.340 | **0.339** | 0.340 | 0.341 | 0.344 | 0.344 | 0.358 | 0.349 | 0.351 | 0.348 | 0.365 | 0.359 | 0.354 | 0.340 | 0.398 | 0.418 | 0.345 | 0.381 |
| | Avg | **0.239** | 0.271 | 0.240 | **0.267** | 0.241 | 0.271 | 0.243 | 0.271 | 0.258 | 0.278 | 0.249 | 0.278 | 0.259 | 0.287 | 0.259 | 0.273 | 0.259 | 0.315 | 0.265 | 0.317 |
| PEMS03 | 12 | **0.062** | **0.165** | 0.070 | 0.178 | 0.070 | 0.173 | 0.066 | 0.172 | 0.071 | 0.174 | 0.078 | 0.187 | 0.085 | 0.192 | 0.099 | 0.216 | 0.090 | 0.203 | 0.122 | 0.243 |
| | 24 | **0.079** | **0.188** | 0.101 | 0.213 | 0.092 | 0.194 | 0.089 | 0.201 | 0.093 | 0.201 | 0.108 | 0.218 | 0.118 | 0.223 | 0.142 | 0.259 | 0.121 | 0.240 | 0.201 | 0.317 |
| | 48 | **0.111** | **0.221** | 0.162 | 0.272 | 0.129 | 0.229 | 0.136 | 0.247 | 0.125 | 0.236 | 0.178 | 0.272 | 0.155 | 0.260 | 0.211 | 0.319 | 0.202 | 0.317 | 0.333 | 0.425 |
| | 96 | 0.160 | **0.263** | 0.239 | 0.336 | **0.157** | 0.261 | 0.182 | 0.282 | 0.164 | 0.275 | 0.238 | 0.328 | 0.228 | 0.317 | 0.269 | 0.370 | 0.262 | 0.367 | 0.457 | 0.515 |
| | Avg | **0.103** | **0.209** | 0.143 | 0.250 | 0.112 | 0.214 | 0.118 | 0.226 | 0.113 | 0.222 | 0.150 | 0.251 | 0.147 | 0.248 | 0.180 | 0.291 | 0.169 | 0.282 | 0.278 | 0.375 |
| PEMS04 | 12 | **0.071** | **0.172** | 0.088 | 0.197 | 0.074 | 0.178 | 0.078 | 0.186 | 0.078 | 0.183 | 0.086 | 0.199 | 0.087 | 0.195 | 0.105 | 0.224 | 0.098 | 0.218 | 0.148 | 0.272 |
| | 24 | **0.084** | 0.191 | 0.114 | 0.226 | 0.087 | **0.181** | 0.099 | 0.212 | 0.095 | 0.205 | 0.101 | 0.218 | 0.103 | 0.215 | 0.153 | 0.275 | 0.131 | 0.256 | 0.224 | 0.340 |
| | 48 | **0.107** | 0.218 | 0.178 | 0.286 | 0.110 | 0.214 | 0.133 | 0.248 | 0.120 | 0.233 | 0.127 | 0.247 | 0.136 | 0.250 | 0.229 | 0.339 | 0.205 | 0.326 | 0.355 | 0.437 |
| | 96 | 0.155 | 0.258 | 0.252 | 0.347 | **0.148** | 0.251 | 0.167 | 0.281 | 0.150 | 0.262 | 0.174 | 0.292 | 0.190 | 0.303 | 0.291 | 0.389 | 0.402 | 0.457 | 0.452 | 0.504 |
| | Avg | **0.104** | 0.210 | 0.158 | 0.264 | 0.105 | 0.209 | 0.119 | 0.232 | 0.111 | 0.221 | 0.122 | 0.239 | 0.129 | 0.241 | 0.195 | 0.307 | 0.209 | 0.314 | 0.295 | 0.388 |
| PEMS07 | 12 | **0.056** | **0.151** | 0.064 | 0.164 | 0.057 | 0.152 | 0.062 | 0.162 | 0.067 | 0.165 | 0.079 | 0.182 | 0.082 | 0.181 | 0.095 | 0.207 | 0.094 | 0.200 | 0.115 | 0.242 |
| | 24 | **0.073** | **0.172** | 0.091 | 0.196 | 0.079 | 0.179 | 0.086 | 0.192 | 0.088 | 0.190 | 0.099 | 0.206 | 0.101 | 0.204 | 0.150 | 0.262 | 0.139 | 0.247 | 0.210 | 0.329 |
| | 48 | **0.099** | 0.200 | 0.141 | 0.247 | 0.099 | **0.191** | 0.128 | 0.234 | 0.110 | 0.215 | 0.133 | 0.239 | 0.134 | 0.238 | 0.253 | 0.340 | 0.311 | 0.369 | 0.398 | 0.458 |
| | 96 | 0.146 | 0.237 | 0.202 | 0.297 | **0.107** | **0.205** | 0.176 | 0.268 | 0.139 | 0.245 | 0.179 | 0.279 | 0.181 | 0.279 | 0.346 | 0.404 | 0.396 | 0.442 | 0.594 | 0.553 |
| | Avg | 0.094 | 0.190 | 0.125 | 0.226 | **0.085** | **0.182** | 0.113 | 0.214 | 0.101 | 0.204 | 0.122 | 0.227 | 0.125 | 0.226 | 0.211 | 0.303 | 0.235 | 0.315 | 0.329 | 0.396 |
| PEMS08 | 12 | **0.075** | 0.176 | 0.085 | 0.190 | **0.075** | 0.176 | 0.082 | 0.185 | 0.079 | 0.182 | 0.105 | 0.211 | 0.112 | 0.212 | 0.168 | 0.232 | 0.165 | 0.214 | 0.154 | 0.276 |
| | 24 | 0.104 | **0.197** | 0.118 | 0.225 | **0.102** | 0.201 | 0.117 | 0.226 | 0.115 | 0.219 | 0.141 | 0.243 | 0.141 | 0.238 | 0.224 | 0.281 | 0.215 | 0.260 | 0.248 | 0.353 |
| | 48 | **0.141** | **0.229** | 0.204 | 0.290 | 0.158 | 0.248 | 0.169 | 0.268 | 0.186 | 0.235 | 0.211 | 0.300 | 0.198 | 0.283 | 0.321 | 0.354 | 0.315 | 0.355 | 0.440 | 0.470 |
| | 96 | 0.233 | 0.278 | 0.342 | 0.360 | 0.366 | 0.377 | 0.233 | 0.306 | **0.221** | **0.267** | 0.364 | 0.387 | 0.320 | 0.351 | 0.408 | 0.417 | 0.377 | 0.397 | 0.674 | 0.565 |
| | Avg | **0.138** | **0.220** | 0.187 | 0.266 | 0.175 | 0.250 | 0.150 | 0.246 | 0.150 | 0.226 | 0.205 | 0.285 | 0.193 | 0.271 | 0.280 | 0.321 | 0.268 | 0.307 | 0.379 | 0.416 |
| 1st Count | | **47** | **40** | 2 | 5 | 8 | 7 | 1 | 4 | 6 | 5 | 0 | 0 | 0 | 0 | 0 | 0 | 0 | 0 | 0 | 0 |

*Table 13.* Detailed evaluation of plug-and-play capability. This table presents the specific results corresponding to Table 2. 'Prom.' denotes the relative improvement based on the average results (Avg) across all horizons. Best results are highlighted in **bold**.

| | | iTransformer | | | | PatchTST | | | | TimesNet | | | | DLinear | | | |
| | | Original | | +PULSE | | Original | | +PULSE | | Original | | +PULSE | | Original | | +PULSE | |
| Data | H | MSE | MAE | MSE | MAE | MSE | MAE | MSE | MAE | MSE | MAE | MSE | MAE | MSE | MAE | MSE | MAE |
|---|---|---|---|---|---|---|---|---|---|---|---|---|---|---|---|---|---|
| **ETTh1** | 96 | 0.386 | 0.405 | **0.364** | **0.389** | 0.414 | 0.419 | **0.398** | **0.408** | 0.384 | **0.402** | **0.380** | 0.406 | 0.386 | 0.400 | **0.376** | **0.389** |
| | 192 | 0.441 | 0.436 | **0.402** | **0.415** | 0.460 | 0.445 | **0.431** | **0.435** | 0.436 | **0.429** | **0.407** | 0.430 | 0.437 | 0.432 | **0.420** | **0.418** |
| | 336 | 0.487 | 0.458 | **0.428** | **0.434** | 0.501 | 0.466 | **0.449** | **0.459** | 0.491 | 0.515 | **0.448** | 0.456 | 0.481 | 0.459 | **0.455** | **0.434** |
| | 720 | 0.503 | 0.491 | **0.441** | **0.451** | 0.500 | **0.488** | **0.480** | 0.489 | 0.521 | 0.558 | **0.451** | **0.424** | 0.519 | 0.516 | **0.461** | **0.455** |
| | *Avg* | 0.454 | 0.447 | **0.409** | **0.422** | 0.469 | 0.454 | **0.440** | **0.448** | 0.458 | 0.507 | **0.422** | **0.429** | 0.456 | 0.452 | **0.428** | **0.424** |
| | Prom. | – | – | 9.9% | 5.6% | – | – | 6.2% | 1.3% | – | – | 7.9% | 15.4% | – | – | 6.1% | 6.2% |
| **ETTh2** | 96 | 0.297 | 0.349 | **0.274** | **0.332** | 0.302 | 0.348 | **0.291** | **0.346** | 0.340 | 0.374 | **0.295** | **0.347** | 0.333 | 0.387 | **0.290** | **0.335** |
| | 192 | 0.380 | 0.400 | **0.365** | **0.388** | 0.388 | 0.400 | **0.369** | **0.392** | 0.402 | 0.414 | **0.380** | **0.398** | 0.477 | 0.467 | **0.372** | **0.388** |
| | 336 | 0.428 | 0.432 | **0.416** | **0.428** | 0.426 | 0.433 | **0.415** | **0.430** | 0.452 | 0.452 | **0.424** | **0.439** | 0.594 | 0.541 | **0.408** | **0.421** |
| | 720 | **0.427** | **0.445** | 0.432 | 0.448 | 0.431 | 0.446 | **0.423** | **0.445** | 0.462 | 0.468 | **0.432** | **0.452** | 0.831 | 0.657 | **0.421** | **0.440** |
| | *Avg* | 0.383 | 0.407 | **0.372** | **0.399** | 0.387 | 0.407 | **0.375** | **0.403** | 0.414 | 0.427 | **0.383** | **0.409** | 0.559 | 0.515 | **0.373** | **0.396** |
| | Prom. | – | – | 2.9% | 2.0% | – | – | 3.1% | 1.0% | – | – | 7.5% | 4.2% | – | – | 33.3% | 23.1% |
| **ETTm1** | 96 | 0.386 | 0.405 | **0.300** | **0.342** | 0.329 | 0.367 | **0.313** | **0.359** | 0.338 | 0.375 | **0.319** | **0.362** | 0.345 | 0.372 | **0.314** | **0.354** |
| | 192 | 0.441 | 0.436 | **0.344** | **0.371** | 0.367 | **0.385** | 0.354 | 0.387 | 0.374 | 0.387 | **0.360** | **0.386** | 0.380 | 0.389 | **0.363** | **0.386** |
| | 336 | 0.487 | 0.458 | **0.372** | **0.392** | 0.399 | 0.410 | **0.378** | **0.405** | 0.410 | 0.411 | **0.378** | **0.399** | 0.413 | 0.413 | **0.389** | **0.402** |
| | 720 | 0.503 | 0.491 | **0.430** | **0.428** | 0.454 | 0.439 | **0.423** | **0.433** | 0.478 | 0.450 | **0.432** | **0.439** | 0.474 | 0.453 | **0.439** | **0.433** |
| | *Avg* | 0.407 | 0.410 | **0.362** | **0.383** | 0.387 | 0.400 | **0.367** | **0.396** | 0.400 | 0.406 | **0.372** | **0.397** | 0.403 | 0.407 | **0.376** | **0.394** |
| | Prom. | – | – | 11.1% | 6.6% | – | – | 5.2% | 1.0% | – | – | 7.0% | 2.2% | – | – | 6.7% | 3.2% |
| **ETTm2** | 96 | 0.180 | 0.264 | **0.165** | **0.248** | 0.175 | 0.259 | **0.164** | **0.249** | 0.187 | 0.267 | **0.176** | **0.263** | 0.193 | 0.292 | **0.170** | **0.255** |
| | 192 | 0.250 | 0.309 | **0.229** | **0.291** | 0.241 | 0.302 | **0.231** | **0.297** | 0.249 | 0.309 | **0.200** | **0.271** | 0.284 | 0.362 | **0.235** | **0.299** |
| | 336 | 0.311 | 0.348 | **0.286** | **0.329** | 0.305 | 0.343 | **0.289** | **0.335** | 0.321 | 0.351 | **0.304** | **0.343** | 0.369 | 0.427 | **0.295** | **0.338** |
| | 720 | 0.412 | 0.407 | **0.395** | **0.394** | 0.402 | 0.400 | **0.392** | **0.395** | 0.408 | 0.403 | **0.395** | **0.396** | 0.554 | 0.522 | **0.396** | **0.395** |
| | *Avg* | 0.288 | 0.332 | **0.269** | **0.316** | 0.281 | 0.326 | **0.269** | **0.319** | 0.291 | 0.333 | **0.269** | **0.318** | 0.350 | 0.401 | **0.274** | **0.322** |
| | Prom. | – | – | 6.6% | 4.8% | – | – | 4.3% | 2.1% | – | – | 7.6% | 4.5% | – | – | 21.7% | 19.7% |
| **Electricity** | 96 | 0.148 | 0.240 | **0.129** | **0.224** | 0.181 | 0.270 | **0.156** | **0.263** | 0.168 | 0.272 | **0.158** | **0.262** | 0.197 | 0.282 | **0.141** | **0.236** |
| | 192 | 0.162 | **0.253** | 0.156 | 0.254 | 0.188 | 0.274 | **0.165** | **0.265** | 0.184 | 0.289 | **0.171** | **0.275** | 0.196 | 0.285 | **0.160** | **0.256** |
| | 336 | 0.178 | 0.269 | **0.158** | **0.258** | 0.204 | 0.293 | **0.182** | **0.283** | 0.198 | 0.300 | **0.185** | **0.290** | 0.209 | 0.301 | **0.175** | **0.275** |
| | 720 | 0.225 | 0.317 | **0.192** | **0.295** | 0.246 | 0.324 | **0.200** | **0.304** | 0.220 | **0.320** | 0.213 | 0.322 | 0.245 | 0.333 | **0.197** | **0.299** |
| | *Avg* | 0.178 | 0.270 | **0.159** | **0.258** | 0.205 | 0.290 | **0.176** | **0.279** | 0.192 | 0.295 | **0.182** | **0.287** | 0.212 | 0.300 | **0.168** | **0.267** |
| | Prom. | – | – | 10.7% | 4.4% | – | – | 14.1% | 3.8% | – | – | 5.2% | 2.7% | – | – | 20.8% | 11.0% |
| **Traffic** | 96 | 0.395 | 0.268 | **0.389** | **0.253** | **0.462** | 0.295 | 0.523 | 0.294 | 0.593 | 0.321 | **0.521** | **0.297** | 0.650 | 0.396 | **0.504** | **0.300** |
| | 192 | 0.417 | 0.276 | **0.413** | **0.264** | **0.466** | 0.296 | 0.541 | 0.291 | 0.617 | 0.336 | **0.535** | **0.302** | 0.598 | 0.370 | **0.504** | **0.307** |
| | 336 | **0.433** | 0.283 | 0.433 | **0.275** | **0.482** | 0.304 | 0.584 | 0.300 | 0.629 | **0.336** | 0.591 | 0.354 | 0.605 | 0.373 | **0.512** | **0.314** |
| | 720 | 0.467 | 0.302 | **0.461** | **0.295** | **0.514** | 0.322 | 0.610 | 0.318 | 0.640 | 0.350 | **0.565** | **0.324** | 0.645 | 0.394 | **0.541** | **0.331** |
| | *Avg* | 0.428 | 0.282 | **0.424** | **0.272** | **0.481** | 0.304 | 0.565 | 0.301 | 0.620 | 0.336 | **0.553** | **0.319** | 0.625 | 0.383 | **0.515** | **0.313** |
| | Prom. | – | – | 0.9% | 3.5% | – | – | -17.5% | 1.0% | – | – | 10.8% | 5.1% | – | – | 17.6% | 18.3% |
| **Weather** | 96 | 0.174 | 0.241 | **0.145** | **0.191** | 0.177 | 0.218 | **0.152** | **0.198** | 0.172 | 0.220 | **0.159** | **0.212** | 0.196 | 0.255 | **0.164** | **0.216** |
| | 192 | 0.221 | **0.254** | 0.204 | 0.254 | 0.225 | 0.259 | **0.208** | **0.252** | 0.219 | **0.261** | **0.211** | 0.263 | 0.237 | 0.296 | **0.210** | **0.258** |
| | 336 | 0.278 | **0.296** | 0.264 | 0.301 | 0.278 | 0.297 | **0.263** | **0.294** | 0.280 | **0.306** | **0.227** | 0.310 | 0.283 | 0.335 | **0.269** | **0.301** |
| | 720 | 0.358 | **0.347** | 0.334 | 0.352 | 0.354 | **0.348** | 0.345 | 0.350 | 0.365 | 0.359 | **0.338** | 0.355 | **0.345** | 0.381 | 0.357 | **0.364** |
| | *Avg* | 0.258 | 0.282 | **0.237** | **0.275** | 0.259 | 0.281 | **0.242** | **0.274** | 0.259 | 0.287 | **0.234** | **0.285** | 0.265 | 0.317 | **0.250** | **0.285** |
| | Prom. | – | – | 8.1% | 2.5% | – | – | 6.6% | 2.5% | – | – | 9.7% | 0.7% | – | – | 5.7% | 10.1% |
| **Solar** | 96 | 0.203 | **0.237** | 0.166 | 0.240 | 0.234 | 0.286 | **0.186** | **0.247** | 0.250 | 0.292 | **0.162** | **0.238** | 0.290 | 0.378 | **0.187** | **0.246** |
| | 192 | 0.233 | **0.261** | 0.197 | 0.266 | 0.267 | 0.310 | **0.219** | **0.288** | 0.296 | 0.318 | **0.200** | **0.271** | 0.320 | 0.398 | **0.218** | **0.274** |
| | 336 | 0.248 | 0.273 | **0.200** | **0.260** | 0.290 | 0.315 | **0.221** | **0.283** | 0.319 | 0.330 | **0.239** | **0.291** | 0.353 | 0.415 | **0.225** | **0.285** |
| | 720 | 0.249 | 0.275 | **0.208** | **0.262** | 0.289 | 0.317 | **0.239** | **0.301** | 0.338 | 0.337 | **0.210** | **0.265** | 0.356 | 0.413 | **0.235** | **0.281** |
| | *Avg* | 0.233 | 0.262 | **0.193** | **0.257** | 0.270 | 0.307 | **0.216** | **0.280** | 0.301 | 0.319 | **0.203** | **0.266** | 0.330 | 0.401 | **0.216** | **0.272** |
| | Prom. | – | – | 17.2% | 1.9% | – | – | 20.0% | 8.8% | – | – | 32.6% | 16.6% | – | – | 34.5% | 32.2% |

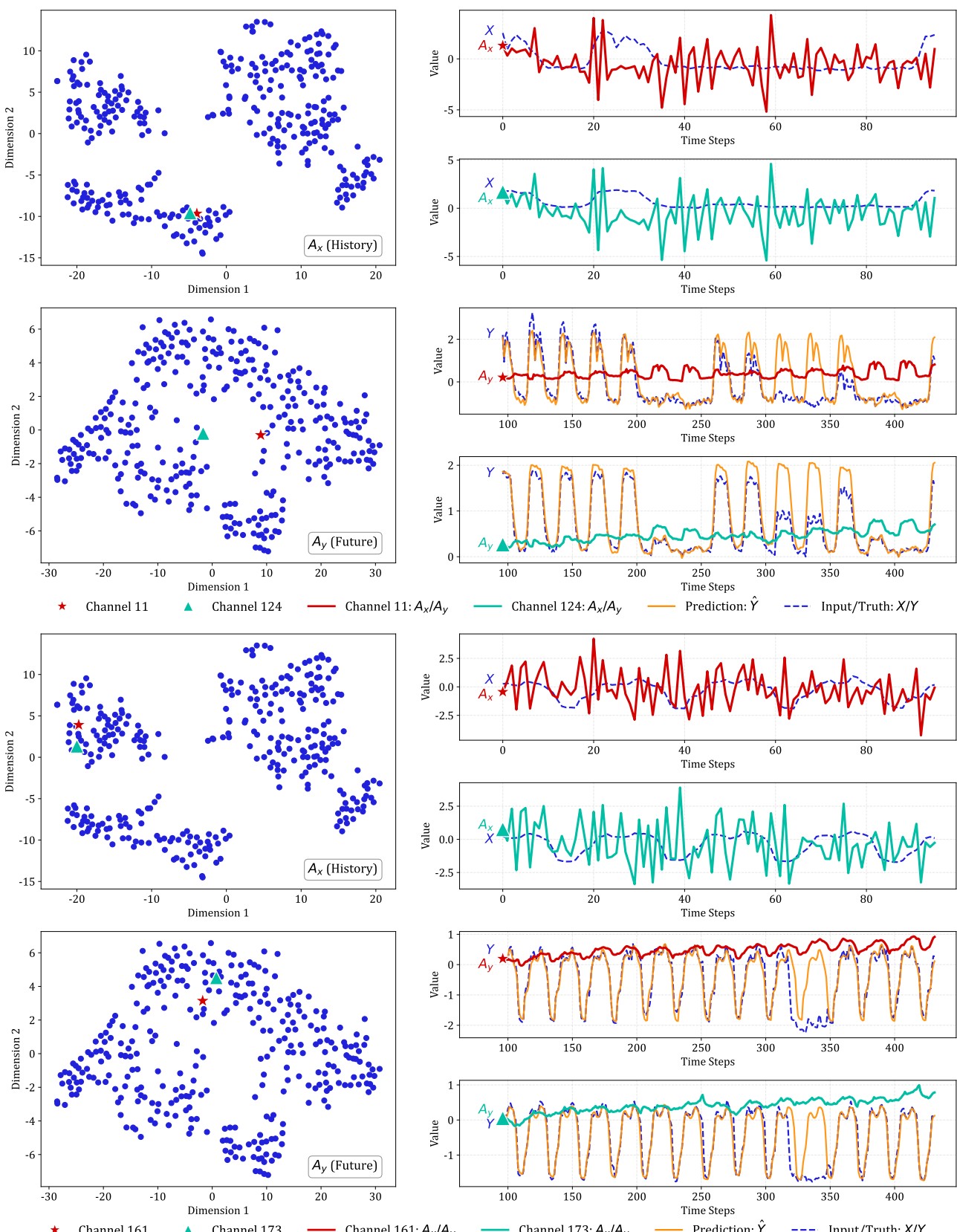

*Figure 10.* Visual analysis of learned Phase Anchors versus raw data on the Electricity dataset via T-SNE. The visualization illustrates that PULSE successfully captures the underlying structural similarities between distinct channels, effectively mapping input sequences $X$ and their corresponding future ground truth $Y$ into a coherent representation space.

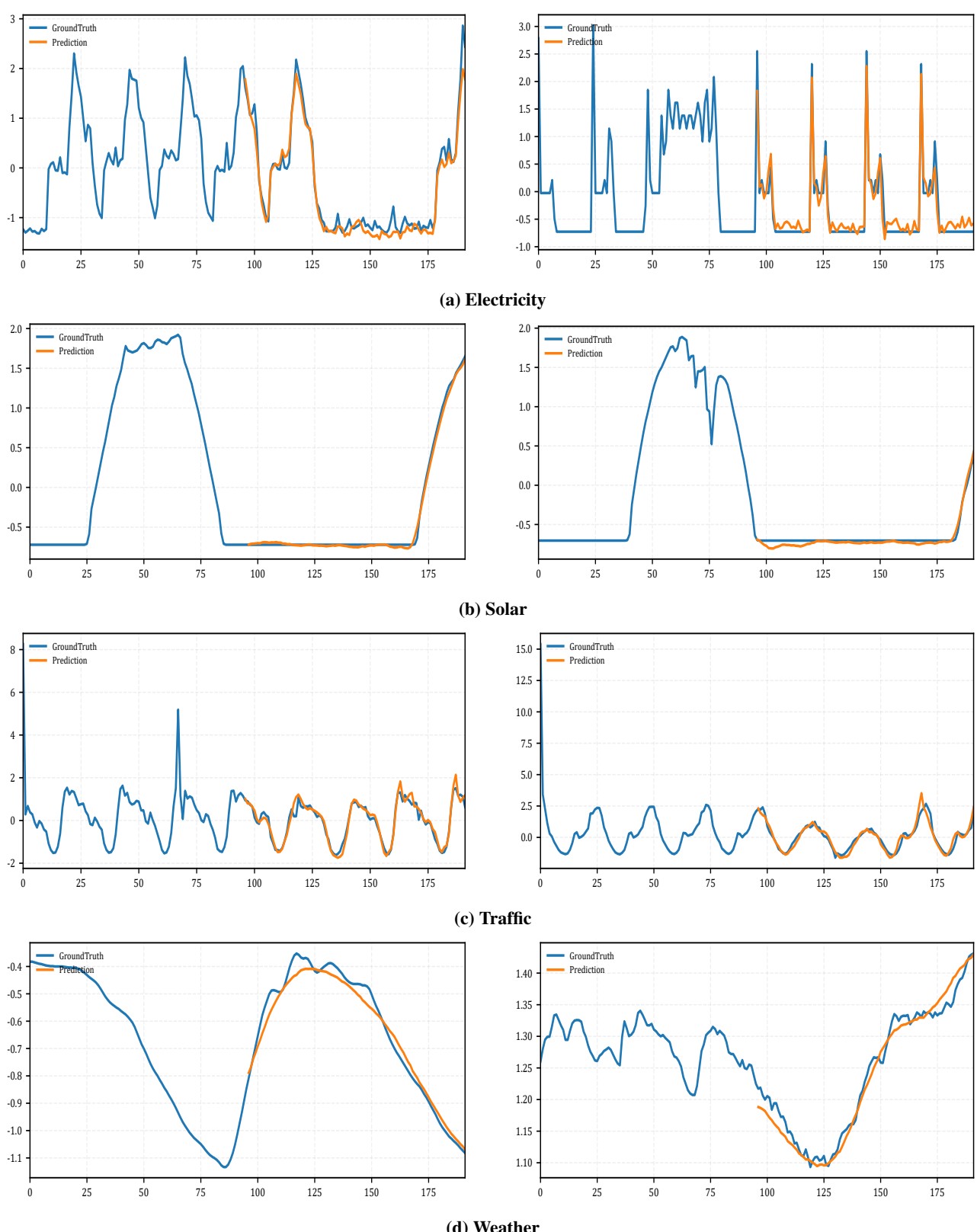

*Figure 11.* Visualization of the forecasting results on four representative datasets. From top to bottom, the results are shown on Electricity, Solar, Traffic, and Weather. For each dataset, two representative forecasting cases are displayed side by side, comparing the predicted sequence with the corresponding ground truth.

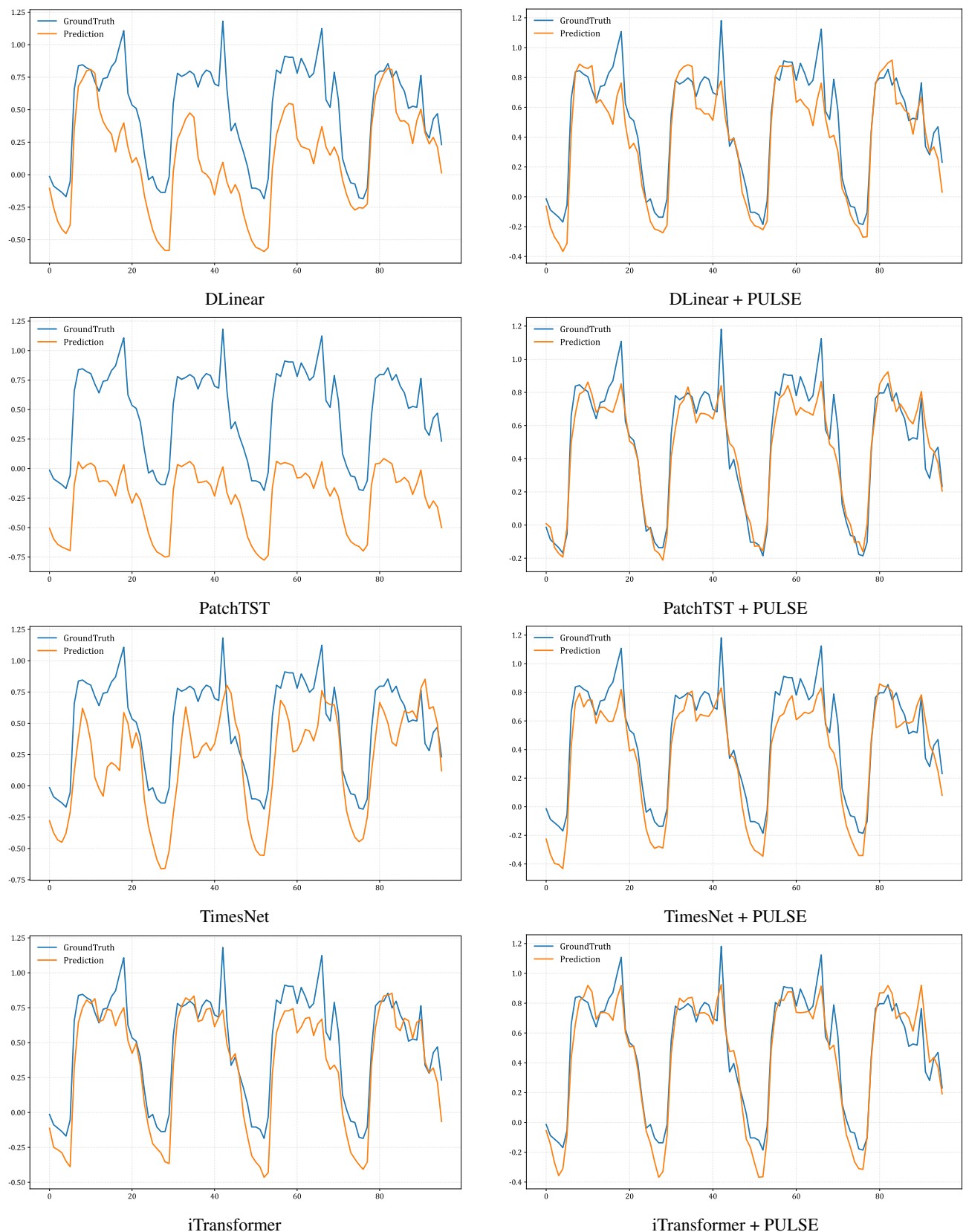

*Figure 12.* Visualization of the plug-and-play forecasting results (input-96-predict-96) on the Electricity dataset. Each row compares the original performance of DLinear, PatchTST, TimesNet, and iTransformer (left) against the enhanced results after integrating our PULSE framework (right). It is evident that PULSE consistently rectifies the "Phase Amnesia" issue, leading to more precise alignment with the ground truth.

