# OpenReview forum: "PULSE: Generative Phase Evolution for Non-Stationary Time Series Forecasting"
_ICML.cc/2026/Conference — ICML 2026 regular_

### Official Review · Reviewer_cfou · 2026-03-01

**Soundness:** 3
**Presentation:** 3
**Significance:** 3
**Originality:** 3
**Overall Recommendation:** 5
**Confidence:** 4

**Summary:**

This paper proposes a new approach for non-stationary multivariate time series forecasting. The key idea is the Disentangle-Evolve-Simulate paradigm, where time series are decoupled into deterministic phase anchors and stochastic residuals. This leads to a plug-and-play framework. The paper also demonstrates state-of-the-art performance on multiple real-world benchmarking datasets.

**Compliance With Llm Reviewing Policy:**

Affirmed.

**Final Justification:**

The authors have provided extra results, which have addressed the concerns in the original review.

**Key Questions For Authors:**

Please refer to Weaknesses 1 & 2. No further questions.

**Limitations:**

Yes

**Strengths And Weaknesses:**

Strengths:

1. The codes are provided in the supp. file and the quality is quite good.
2. Overall, the paper is well-written, and the observation around "Phase Amnesia" makes sense. The authors also articulate the motivation very well in the introduction with a figure and hypotheses.
3. Achieving SOTA results with an MLP backbone shows a valuable direction for the community.

Weaknesses:
1. The paper claims marginal computational overhead and focuses on theoretical FLOPs. However, it is not clear how these theoretical reductions would lead to actual improvements on the inference stage.
2. The paper limits the model capacity based on the "Low-Rank Hypothesis". However, there is no ablation showing what happens if $d_{model}$ is scaled up. For example, whether the performance would drop with a larger $d_{model}$ (e.g., $d_{model} = 512$) (line 272, $d_{model} \leq 32$).

---

> ### Author Rebuttal · Authors · 2026-03-29
>
> We sincerely thank the reviewer for the insightful comments. Below, we respond to the remaining questions in turn.
>
> **W1: Practical Inference Overhead beyond Theoretical FLOPs**
>
> Thank you for this important comment. We agree that theoretical FLOPs are only a proxy and do not fully characterize practical inference behavior. To address this, we additionally measure **actual inference time** and **peak GPU memory** in the plug-and-play setting. Specifically, during inference, we select a single batch, run it **100 times**, report the **median inference latency** (T, in ms), and record the **peak GPU memory** (M, in GB).
>
> The results show that PULSE introduces **relatively modest overhead** in practice. For medium-to-heavy backbones (iTransformer, PatchTST, TimesNet), the additional latency is generally **within a few milliseconds** and peak memory increases tend to be **minor**. For the lightweight DLinear, the relatively larger increase is mainly due to its **near-zero baseline cost**, while the absolute overhead remains reasonable. Overall, these results suggest that PULSE can achieve **meaningful forecasting improvements with acceptable runtime and memory costs**.
>
> **The lookback length is fixed at 96, and all metrics are averaged over prediction horizons 96 / 192 / 336 / 720.**
>
> |Dataset|ETTh1|||Electricity|||Weather|||Solar|||
> |:--|:--:|:--:|:--:|:--:|:--:|:--:|:--:|:--:|:--:|:--:|:--:|:--:|
> |Metric|MSE,MAE|T(ms)|M(GB)|MSE,MAE|T(ms)|M(GB)|MSE,MAE|T(ms)|M(GB)|MSE,MAE|T(ms)|M(GB)|
> |iTransformer|0.454,0.447|4.068|0.130|0.178,0.270|20.910|0.904|0.258,0.282|5.029|0.238|0.233,0.262|8.358|0.291|
> |+PULSE|**0.409,0.422**|6.598|0.131|**0.159,0.258**|28.580|0.945|**0.237,0.275**|8.965|0.239|**0.193,0.257**|11.178|0.316|
> |PatchTST|0.469,0.454|8.028|0.415|0.205,0.290|87.483|4.315|0.259,0.281|22.023|1.163|0.270,0.216|38.678|1.868|
> |+PULSE|**0.440,0.448**|9.908|0.422|**0.176,0.279**|94.188|4.364|**0.242,0.274**|25.655|1.175|**0.216,0.280**|40.290|1.889|
> |TimesNet|0.458,0.507|40.848|0.993|0.192,0.295|9.843|0.312|0.259,0.287|39.980|0.999|0.301,0.319|10.138|0.293|
> |+PULSE|**0.422,0.429**|43.750|0.999|**0.182,0.287**|13.888|0.447|**0.234,0.285**|43.158|1.013|**0.203,0.266**|13.040|0.316|
> |DLinear|0.456,0.452|0.438|0.043|0.212,0.300|0.625|0.158|0.265,0.317|0.485|0.064|0.330,0.401|0.490|0.086|
> |+PULSE|**0.428,0.424**|3.828|0.077|**0.168,0.267**|8.010|0.409|**0.250,0.285**|4.615|0.164|**0.216,0.272**|5.490|0.210|
>
> **W2: Model Capacity under the Low-Rank Hypothesis**
>
> Thank you for this important question. The low-rank hypothesis posits that deterministic phase evolution resides on a compact manifold with sparse basis dependencies. Our Phase Tokenization captures this by learning condensed phase relationships, which fundamentally obviates the need for large $d_{\text{model}}$.
>
> Following your suggestion, we conducted ablation studies to empirically validate this hypothesis by scaling $d_{\text{model}}$ from our default (16/32) to 64, 128, 256, and 512. Results show that our default small $d_{\text{model}}$ consistently performs best across all six datasets. Increasing $d_{\text{model}}$ leads to gradual degradation, which is mild at 64/128 but becomes much more evident at 256/512. This demonstrates that excessive capacity disrupts alignment with the intrinsic low-rank phase manifold.
>
> **Effect of scaling up $d_{\text{model}}$. Each metric is averaged over prediction horizons 96/192/336/720.**
>
> | $d_{\text{model}}$| 16/32 |  | 64 |  | 128 |  | 256 |  | 512 |  |
> |:--------|:------|:------|:------|:------|:------|:------|:------|:------|:------|:------|
> | Dataset | MSE | MAE | MSE | MAE | MSE | MAE | MSE | MAE | MSE | MAE |
> | ETTh1 | **0.412** | **0.422** | 0.420 | 0.425 | 0.421 | 0.426 | 0.426 | 0.426 | 0.593 | 0.509 |
> | ETTh2 | **0.361** | **0.391** | 0.436 | 0.395 | 0.370 | 0.397 | 0.377 | 0.401 | 0.416 | 0.432 |
> | ETTm1 | **0.363** | **0.388** | 0.368 | 0.390 | 0.368 | 0.391 | 0.401 | 0.404 | 0.435 | 0.425 |
> | ETTm2 | **0.262** | **0.314** | 0.268 | 0.317 | 0.268 | 0.317 | 0.279 | 0.325 | 0.297 | 0.334 |
> | Electricity | **0.160** | **0.255** | 0.164 | 0.257 | 0.164 | 0.257 | 0.185 | 0.279 | 0.203 | 0.298 |
> | Weather | **0.239** | **0.271** | 0.242 | 0.274 | 0.243 | 0.276 | 0.274 | 0.304 | 0.279 | 0.309 |

---

> > ### Author Rebuttal · Reviewer_cfou · 2026-04-03
> >
> > Thank you for the extra results. They have resolved the concerns.

---

> > > ### Author Response · Authors · 2026-04-04
> > >
> > > We sincerely thank the reviewer for the careful follow-up and for taking the time to evaluate the additional experimental results. We are glad that the added analysis on practical inference overhead and model capacity under the low-rank hypothesis helped resolve the concerns.
> > >
> > > We truly appreciate the reviewer’s positive feedback and the decision to increase the score. It is encouraging to see that our clarifications and additional experiments improved the understanding of the proposed method.
> > >
> > > Thank you again for your constructive feedback and support.

---

### Official Review · Reviewer_3rAW · 2026-03-10

**Soundness:** 3
**Presentation:** 3
**Significance:** 3
**Originality:** 3
**Overall Recommendation:** 5
**Confidence:** 5

**Summary:**

The paper proposes PULSE, a framework to tackle "Phase Amnesia" in non-stationary time series forecasting. It introduces a "Disentangle–Evolve–Simulate" paradigm comprising three key modules: Phase-Anchored Disentanglement, a Generative Phase Router, and Statistic-Aware Mixup (SAM). The authors hypothesize that non-stationarity should be modeled by evolving the deterministic phase trajectory rather than passive historical lookup. Empirically, PULSE achieves state-of-the-art (SOTA) results on 12 datasets, notably boosting simple MLP backbones to outperform complex Transformers.

**Compliance With Llm Reviewing Policy:**

Affirmed.

**Final Justification:**

Thank you for the detailed rebuttal. The responses addressed my main concerns clearly and improved my confidence in the paper.

Most importantly, the clarification that the Phase Router is better understood as a **generative manifold learner** rather than a rigid phase lookup mechanism helps reconcile the model’s robustness under misaligned \(W\) with the paper’s physical motivation. I also appreciate the authors’ acknowledgment that the original CI-based explanation for the Traffic result was insufficient. The revised interpretation in terms of **inductive bias mismatch** between PatchTST’s patch aggregation and PULSE’s disentangled residual stream is much more convincing. In addition, the explanation that \(Y_0\) acts only as a conditioning/query signal, rather than directly injecting stochastic noise into the anchor, makes the information flow in Eq. (4) more reasonable. Finally, the tuning discussion suggests that although \(L\) matters on some datasets, the practical search space remains small. Overall, I am satisfied with the rebuttal and will raise my score.

**Key Questions For Authors:**

Based on the Weaknesses, I have the following questions:
1.  **Mechanism of Robustness:** Can you theoretically explain *why* the Phase Router remains effective even when the global period W is misaligned? Is it performing some form of dynamic error correction?
2.  **Traffic Dataset Explanation:** Why specifically does PatchTST fail with PULSE on Traffic data, while the simpler MLP succeeds, given both are CI? Is there a conflict between Patching and your residual disentanglement?
3.  **Router Information Flow:** Please clarify the intuition behind using the residual prediction Y0 to drive the phase evolution. How do you ensure the generated anchor Ay remains smooth if it is derived from noise?
4.  **Tuning Cost:** For datasets like Solar, does the sensitivity to L imply a high tuning overhead? How does this align with the "Low-Rank" hypothesis?

**Limitations:**

Yes

**Strengths And Weaknesses:**

**Strengths:**

* **Novel Conceptualization:** The shift from "passive fitting" to "generative evolution" is refreshing. The theoretical formalization of "Phase Amnesia" provides a strong motivation.
* **Efficiency:** Achieving SOTA performance with a lightweight MLP backbone suggests the proposed inductive biases are highly effective. The linear complexity  analysis is compelling.
* **Robustness Strategy:** The theoretical analysis of Mixup stability and the proposed SAM method offer a grounded solution to scale collapse.

**Weaknesses:**

1. **Contradiction in Robustness:** The model claims to rely on a physical period , yet Figure 5 shows it works well even when  is wrong. This empirical robustness contradicts the physical hypothesis. Is the Phase Router actually learning physics, or just overfitting to a scrambled latent space?
2. **Inconsistent Failure on Traffic:** The paper attributes the failure of "PatchTST + PULSE" to PatchTST's Channel-Independent (CI) nature. However, the default MLP backbone is also CI but performs remarkably well. This explanation is logically flawed.
3. **Risk of Instability:** Equation (4) uses the stochastic residual  to guide the deterministic anchor . Conditioning a stable trend on high-entropy noise seems theoretically risky and circular.
4. **Tuning Sensitivity:** Figure 6 shows high sensitivity to Codebook Size  on physics-driven datasets. Does this imply a high tuning cost for new applications?

---

> ### Author Rebuttal · Authors · 2026-03-26
>
> We sincerely thank the reviewer for the rigorous and constructive comments. Below, we respond to the questions on robustness, the inductive bias mismatch on Traffic, information flow stability, and tuning costs.
>
> **W1 & Q1: Mechanism of Robustness under Misaligned Period**
>
> The empirical robustness does not contradict the physical hypothesis. Rather, it suggests that the Phase Router functions as a **generative manifold learner** rather than a rigid historical lookup table.
>
> - **Scrambled state as a usable structural basis.** Even when the global period $W$ is not perfectly aligned, the time index and codebook $L$ still provide a deterministic structural basis in latent space.
> - **Robustness to moderate misalignment.** The Phase Router does not rely on exact index matching. Instead, the cross-attention in Eq. (4) learns a flexible mapping conditioned on $Y_0$, making the model robust to moderate phase misalignment. As long as the anchor avoids degenerate structure ($W > 1$), the router can still generate meaningful trajectories. In this sense, it evolves the deterministic anchor under residual context, rather than rigid lookup or exact phase recovery.
>
> **W2 & Q2: Inconsistent Failure on Traffic**
>
> We thank you for this observation. Your point is well taken. Since our default MLP backbone is also Channel-Independent (CI), attributing the failure solely to PatchTST's CI nature is not fully justified. A more plausible explanation is an inductive bias mismatch, and we will revise this explanation accordingly.
>
> - **Residual characteristics after disentanglement.** PULSE moves dominant structural dynamics into the anchor stream, leaving the residual stream $R$ more stochastic and less locally smooth in time.
> - **The conflict with patching.** PatchTST assumes short temporal neighborhoods contain stable local semantics that can be aggregated into useful patch-level representations. This bias becomes less suitable for PULSE's residual stream, where local smoothness is weakened. On Traffic, the mismatch is more severe because dense sensor interactions make fine-grained dependencies harder to preserve during patch-level aggregation.
> - **Why the MLP succeeds.** In contrast, the point-wise MLP makes weaker assumptions about local temporal smoothness and can model the residual directly as a stochastic signal, which is more compatible with our Statistic-Aware Mixup (SAM) design.
> - **What the current evidence supports.** PatchTST + PULSE drops on Traffic in the plug-and-play analysis, whereas other backbones improve. Meanwhile, PULSE regains top-1 performance on PEMS08 in the main results. This suggests that the Traffic failure is better explained by a dataset-specific inductive bias mismatch, rather than by CI alone or an inherent incompatibility with PatchTST.
>
> **W3 & Q3: Router Information Flow and Stability from $Y_0$**
>
> Conditioning the deterministic anchor on the residual prediction $Y_0$ is not circular. Instead, $Y_0$ provides complementary information about the recent stochastic state:
>
> - **Residual as a modulation signal.** In Eq. (4), $Y_0$ serves as a latent query that informs how the deterministic trajectory should evolve under the current residual context. It captures recent volatility regime and trend-deviation signals.
> - **Why this does not destabilize $A_y$.** The cross-attention uses $Y_0$ as a conditioning/query signal to modulate anchor evolution based on $A_x$, rather than directly copying $Y_0$'s structure. The anchor prior $A_x$ preserves the smoothness and structural consistency of $A_y$. Thus, $Y_0$ provides directional information without directly injecting raw stochastic noise. This is also consistent with our design: residuals may be less locally smooth in time, while still carrying structured statistical information for modulation.
>
> **W4 & Q4: Tuning Sensitivity and the Low-Rank Hypothesis**
>
> Sensitivity to codebook size $L$ on datasets such as Solar does not necessarily imply high tuning overhead. It only means structural resolution matters more on these datasets, not that $L$ must be searched over a large space.
>
> - **Why sensitivity can be higher.** $L$ controls phase codebook resolution. For datasets with clearer periodic structure, structural discretization matters more, so stronger dependence on $L$ is expected.
> - **Why tuning overhead remains limited.** $L$ is selected from only $\{12, 24, 32, 48, 96\}$, a small discrete set. Figure 6 also shows wide plateaus, indicating that approximate tuning is often sufficient. Domain knowledge (e.g., 24 for hourly data) further narrows the candidate range.
> - **How this aligns with the Low-Rank Hypothesis.** The Low-Rank Hypothesis does not imply complete insensitivity to $L$. Rather, it suggests that the effective structural manifold has limited resolution, so the useful search space remains small and controllable. Thus, the observed sensitivity on Solar is consistent with low-rank structure, while the overall tuning cost remains modest.

---

> > ### Author Rebuttal · Reviewer_3rAW · 2026-04-03
> >
> > Thank you for your detailed response. I will raise my score.

---

> > > ### Author Response · Authors · 2026-04-04
> > >
> > > We sincerely thank the reviewer for the thoughtful follow-up and for taking the time to carefully consider our rebuttal. We are especially grateful that our clarifications regarding robustness, inductive bias mismatch, information flow, and tuning sensitivity helped address the concerns.
> > >
> > > We truly appreciate the reviewer’s decision to raise the score, and we are encouraged that our explanations have clarified the design and empirical behavior of the proposed method.
> > >
> > > Thank you again for your constructive feedback and support.

---

### Official Review · Reviewer_2TZh · 2026-03-11

**Soundness:** 2
**Presentation:** 2
**Significance:** 3
**Originality:** 3
**Overall Recommendation:** 3
**Confidence:** 4

**Summary:**

This work proposes a three-part framework for improving transformer architectures on non-stationary time-series forecasting tasks. The central motivation is to counteract "Phase Amnesia" which the authors coin to describe a failure mode that arises when the model is blind to evolving global context. Specifically, they claim that many methods exhibit this failure mode on non-stationary data in one of two ways:
- 1) *Scale Collapse*: standardization implicitly assumes the context window is long enough for its statistics to capture the future statistics; and
- 2) *Rigid Periodicity*: the model captures the periodicity/seasonality from a portion of the context and propagates that structure into the forecasts, potentially leading to forecasts being out of phase with the ground truth.

The authors present three hypotheses, which each motivates a stage in their 3-stage method (PULSE):
- 1) *Wold Decomposition Principle*: time-series (including non-stationary time-series) can be decomposed into a deterministic structural component and a stochastic residual component. The authors propose  *Phase Anchor* to implement this splitting.
- 2) *Dynamical Phase Evolution*: real-world signals are often amplitude-modulated and have local phase shifts. Rather than simple historical lookup, a forecaster should learn the underlying time-evolution operator of the dynamics for the deterministic structural component. The authors propose *Phase Router* to model this structural evolution.
- 3) *Heteroscedastic Residual Manifold*:  The stochastic residual component requires treatment under a distributional generalization framework, rather than passive noise fitting. The authors propose *Statistic-Aware Mixup (SAM)* to remix the prediction of the stochastic residual with a synthetic statistic from interpolating the decoding statistics.

The paper evaluates the proposed method, PULSE, on a variety of standard time-series datasets, showing improvement both qualitatively and quantitatively (through two metrics, MSE and MAE). In these evaluations, PULSE is shown as a method with minimal complexity (simple MLP + conv1D + linear adapter), and a plug-and-play addition to existing transformer baselines. The authors also provide aggregated metrics under ablations of the different stages, and some thoughts on the limitations of the approach on specific datasets.

**Compliance With Llm Reviewing Policy:**

Affirmed.

**Final Justification:**

While the authors provided many requested results in their rebuttal, I still believe that the evaluation is limited to too few datasets. Also, in my view the explanations and conceptual framework rely on vague language at times, with some dataset-specific claims. I also am unconvinced regarding my stated Weakness 3; the authors provide example forecasts in their additional supplementary material, but only one example shows a noticeable trend component, and none of them show what the forecasts look like for the baselines. Because of this, I am doubting how much of the evaluation dataset actually contains tasks with strong trends, which seem central to the motivation of the proposed method.

And while appreciated, the quantification of the phase amnesia provided in the rebuttal is difficult to interpret, since the authors don't connect it with the advantage of their method (in my reading, I did not see any attempt to investigate whether the proposed method achieves a greater performance gain on the datasets with larger phase amnesia, which would be the most meaningful demonstration of the usefulness of their framework). In my opinion, the whole phase amnesia framework means little if this relationship is not quantified in a large-scale evaluation.

I therefore maintain my score of weak reject. While I recommend against it, I do not mind if this paper is accepted; I defer the final decision to the AC, whom I strongly encourage to consider the points I've raised above.

**Key Questions For Authors:**

**Questions**
- 1) To my understanding, the Wold Decomposition, at least canonically, only applies to covariance-stationary time-series. Could the authors expound on why it is a valid justification when applied to non-stationary time-series?
- 2) Apologies if I missed this, but could the authors clarify if PULSE significantly increases the inference time of the baseline methods when used as a plug-and-play method?
- 3) From the discussion in Section 4.2, would it be accurate to say that PULSE requires multivariate architectures for plug-and-play compatibility?

**Limitations:**

**Limitations**
- 1) Experiments are restricted to short context lengths of 92
- 2) Only MSE and MAE metrics

**Strengths And Weaknesses:**

**Strengths**
- 1) *Parameter Efficiency*: the authors demonstrate that PULSE applied to a simple MLP can outperform many baselines on many datasets in their evaluation setting. This represents a meaningful advantage of a structured ground-up approach, and it is compelling to move away from higher parameter counts.
- 2) *Evaluation across a wide variety of baselines*: The authors evaluate their approach on 12 standard time-series benchmark datasets, and against 9 baseline methods, showing that it consistently outperforms on MSE and MAE on nearly all datasets and prediction lengths, in their evaluation setting. The suite of baseline methods looks appropriate, especially as some, such as PatchTST, are precursors to several modern time-series foundation models
- 3) *Strength of the plug-and-play approach*: In addition to showing the advantage of their minimalistic PULSE on a basic MLP-based architecture, the authors also show the advantage when using PULSE on top of the baseline methods. This shows clear promise, at least for the context length considered in the study.

**Weaknesses**
- 1) *Experiments are only on short context lengths*: My main concern is that the empirical validation relies on forecasting tasks with unusually short context length of 96 timepoints for all tasks, even the long-term datasets, as the authors detail in Section 4.1. While I believe the method leads to improvement on many datasets at short context lengths, I am wholly unconvinced that it will provide significant benefit when using the standard context lengths of at least 512 (more commonly, over 4000 timepoints) that modern time-series foundation models utilize. As such, I believe the lack of empirical verification for longer context lengths significantly limits the applicability of the PULSE approach.
- 2) *Lack of clear demonstration of Phase Amnesia*: Even for the short context length considered in the study, I would like to see larger-scale quantification of the baseline failure mode the authors term Phase Amnesia. From the examples provided, it is not clear to me if this is a significant issue across all datasets, and I presume that longer context length, at the very least, would mitigate the effect of the normalization not capturing fully the future statistics.
- 3) *Lack of example forecasts with clear trends for non-stationarity*: Although the authors present PULSE as a method for non-stationary time-series forecasting, they do not present any examples of forecasts where the ground truth future has a trend. The example forecasts in Figures 3, 9, 10 do not show any clear trend, and look at best weakly non-stationary. On that note, I would appreciate a quantification of non-stationarity within these forecasting tasks.
- 4) *Unclear Benefits of Methodology*: The authors only report two metrics: MSE and MAE between forecast and ground truth future. I believe the line "PULSE achieves the top-1 results in 18 out of 24 metrics" can come across as misleading to readers, as the results are for these two metrics (MSE and MAE), across 12 datasets. I also find the methodology section, section 3, to be difficult to parse. There appears to be some missing notation, such as a missing definition of $\tilde{R}$. While I appreciate the ablation study, I think it would be greatly improved if the authors present what the deterministic and residual stochastic part of the forecasts look like individually. There is also no assessment of the validity of the authors' claim that the deterministic component results from a learned "time-varying evolution operator",  and it is my conviction that this claim, of "learning this continuous evolution operator" is quite a strong claim to make.

---

> ### Author Rebuttal · Authors · 2026-03-28
>
> We thank the reviewer for the constructive comments. Below, we respond to the remaining questions in turn.
>
> **W1 & L1**
>
> The lookback length of 96 follows the standard benchmark setting used in prior studies.Following your suggestion, we additionally test longer lookbacks and plug-and-play performance, both confirming that PULSE remains effective beyond the short-context setting.
>
> **For each dataset, the three rows correspond to lookback lengths 336 / 512 / 720 (top to bottom), and each metric is averaged over prediction horizons 96 / 192 / 336 / 720.**
>
> ||PULSE|CFPT|CycleNet|
> |:--|:--|:--|:--|
> |Dataset|MSE,MAE|MSE,MAE|MSE,MAE|
> |ETTh1|**0.409**,**0.425**|0.423,0.431|0.415,0.426|
> ||**0.410**,**0.427**|0.416,0.432|0.423,0.434|
> ||**0.409**,**0.428**|0.426,0.441|0.430,0.439|
> |ETTh2|**0.339**,**0.384**|0.341,0.385|0.355,0.398|
> ||**0.335**,**0.385**|0.339,0.388|0.347,0.394|
> ||**0.342**,**0.388**|0.347,0.395|0.345,0.394|
> |ETTm1|**0.344**,**0.374**|0.348,0.376|0.355,0.379|
> ||**0.342**,**0.374**|0.352,0.379|0.358,0.382|
> ||**0.342**,**0.377**|0.353,0.381|0.355,0.381|
> |ETTm2|**0.249**,**0.307**|0.260,0.317|0.251,0.309|
> ||**0.248**,**0.308**|0.262,0.319|0.252,0.311|
> ||**0.247**,**0.308**|0.263,0.322|0.249,0.312|
> |Electricity|**0.149**,**0.246**|0.155,0.249|0.158,0.250|
> ||**0.150**,**0.249**|0.155,**0.249**|0.157,0.251|
> ||**0.149**,**0.250**|0.155,**0.250**|0.157,**0.250**|
> |Weather|**0.222**,**0.262**|0.225,**0.262**|0.226,0.266|
> ||**0.220**,**0.260**|0.223,0.262|0.227,0.267|
> ||**0.219**,**0.261**|0.243,0.276|0.224,0.266|
> ---
> **Plug-and-play results on the Electricity dataset with lookback length = 720.**
> |Horizon|96|192|336|720|
> |:--|:--|:--|:--|:--|
> |Metric|MSE,MAE|MSE,MAE|MSE,MAE|MSE,MAE|
> |iTransformer|0.135,0.233|0.155,0.253|0.169,0.267|0.204,0.301|
> |+PULSE|**0.127,0.224**|**0.147,0.250**|**0.161,0.266**|**0.173,0.280**|
> |DLinear|0.141,0.244|0.155,0.258|0.170,0.275|0.209,0.309|
> |+PULSE|**0.129,0.224**|**0.150,0.245**|**0.165,0.268**|**0.181,0.285**|
> |TimesNet|0.202,0.308|0.218,0.322|0.232,0.332|0.299,0.375|
> |+PULSE|**0.178,0.284**|**0.193,0.299**|**0.208,0.320**|**0.240,0.344**|
> |SparseTSF|0.139,0.239|0.155,0.250|0.171,0.265|0.208,0.300|
> |+PULSE|**0.132,0.223**|**0.147,0.240**|**0.158,0.259**|**0.184,0.285**|
> |PatchTST|0.141,0.240|0.156,0.256|0.172,0.267|0.208,0.299|
> |+PULSE|**0.130,0.227**|**0.146,0.243**|**0.163,0.261**|**0.184,0.294**|
>
> **W2**
>
> To quantify Phase Amnesia under the standard short-context setting ($T=96$), we measure history–future mismatch across horizons using mean shift (MS), std shift (SS), and spectral mismatch (SM):
>
> $$\mathrm{MS}=\frac{|\mu_Y-\mu_X|}{|\mu_Y|+|\mu_X|+\varepsilon},\quad
> \mathrm{SS}=\frac{|\sigma_Y-\sigma_X|}{\sigma_Y+\sigma_X+\varepsilon},\quad
> \mathrm{SM}=\frac{1}{2}\sum_f |\tilde{A}_X(f)-\tilde{A}_Y(f)|,$$
>
> where $X,Y$ are the input and future target, and $\tilde{A}_X,\tilde{A}_Y$ are normalized nonnegative spectra. Mismatch is widespread, usually grows with horizon, while our plug-and-play results show that extended context does not fully remove this issue.
>
> |Dataset|H=96 (MS/SS/SM)|H=192|H=336|H=720|
> |:--|:--:|:--:|:--:|:--:|
> |ETTh1|0.332/0.114/0.267|0.339/0.118/0.349|0.357/0.125/0.386|0.379/0.144/0.389|
> |Weather|0.408/0.261/0.234|0.421/0.267/0.245|0.438/0.297/0.304|0.452/0.348/0.351|
> |Electricity|0.363/0.056/0.196|0.347/0.053/0.459|0.361/0.056/0.580|0.410/0.066/0.564|
> |Traffic|0.704/0.108/0.260|0.621/0.085/0.430|0.603/0.081/0.532|0.645/0.086/0.510|
>
> **W3**
>
> Non-stationarity includes shifts in mean, variance, and spectrum beyond visible trends. As quantified in W2, all datasets show substantial non-stationarity. While Figures 3, 9, 10 showcase periodic and variance shifts, we will add examples with clearer trend components in the revised manuscript.
>
> **W4 & L2**
>
> MSE and MAE are the standard metrics in long-term forecasting benchmarks. In the revision, we will additionally report RMSE and MAPE on short-term datasets, revise the metric wording and define $\hat{R}$. Visualizations of the two components are provided at https://anonymous.4open.science/r/image-56C8. Regarding the "time-varying evolution operator" claim, the Phase Router approximates this operator by learning a data-driven mapping from $A_x$ to $A_y$. The cross-attention mechanism implicitly models how historical phase anchors evolve into future ones, capturing temporal dynamics through learned approximation.
>
> **Q1**
>
> While Wold decomposition is classically defined for stationary processes, we use it here as a structural inductive bias, motivated by its non-stationary extension, to disentangle non-stationary series into deterministic structure and stochastic residuals.
>
> **Q2**
>
> We respectfully refer the reviewer to our response to Reviewer cfou’s W1.
>
> **Q3**
>
> Our framework is general and improves both univariate backbones (e.g., PatchTST) and multivariate backbones (e.g., TimesNet). The difference is mainly data-dependent: datasets with dense inter-variable dependencies favor multivariate backbones more.

---

> > ### Author Rebuttal · Reviewer_2TZh · 2026-04-02
> >
> > I thank the authors for a diligent rebuttal that incorporates some new results requested in my review. However, I believe that the empirical evaluation is limited and requires very substantial improvements, along the lines discussed, to meet the standard for acceptance. I also encourage the authors to report scale-independent metrics such as MASE that are robust to outliers, as a much more reliable evaluation than the MAE and MSE provided, in addition to reporting the longer context lengths (which the authors have made some strides toward in their response). And I encourage the authors to expand the evaluation to a much larger set of datasets to ensure the usefulness of their method, and to incorporate the forecast examples with clear trend components in the next revision.
> >
> > In light of the current current limitations, I maintain my score.

---

> > > ### Author Response · Authors · 2026-04-06
> > >
> > > Thank you for the thoughtful follow-up and for recognizing our efforts in the rebuttal. We agree that broader empirical validation is important. At the same time, we would like to note that our current evaluation already follows standard long-term forecasting benchmarks and verifies the effectiveness of PULSE on 12 public datasets. Following your suggestion, we further extended the evaluation to 5 datasets from the normal-scale benchmark [1] and 1 dataset from the large-scale benchmark [2]. The additional results show that PULSE remains highly effective on these extra benchmarks and further supports its robustness and generality beyond the original evaluation. We also provide forecast examples with clear trend components in the anonymous repository: https://anonymous.4open.science/r/forecasting-632E/README.md.
> > >
> > > **Forecasting results on 5 normal-scale datasets (AQShunyi, AQWan, ZafNoo, CzeLan, and Wind) and 1 large-scale dataset (SD), using a look-back window of 96, averaging metrics over prediction horizons 96 / 192 / 336 / 720, and computing MASE with seasonal period m=1.**
> > >
> > > |  | PULSE |  |  | CFPT |  |  | CycleNet |  |  |
> > > | :-- | :--: | :--: | :--: | :--: | :--: | :--: | :--: | :--: | :--: |
> > > | Dataset | MSE | MAE | MASE | MSE | MAE | MASE | MSE | MAE | MASE |
> > > | SD | **1.299** | **0.201** | **1.597** | 1.493 | 0.209 | 1.663 | 1.341 | 0.254 | 2.027 |
> > > | AQShunyi | **0.734** | **0.517** | **3.469** | 0.779 | 0.526 | 3.527 | 0.764 | 0.521 | 3.494 |
> > > | AQWan | **0.841** | **0.510** | **3.516** | 0.887 | 0.516 | 3.554 | 0.868 | 0.512 | 3.526 |
> > > | ZafNoo | **0.564** | **0.461** | **2.558** | 0.566 | 0.462 | 2.563 | 0.567 | 0.462 | 2.563 |
> > > | CzeLan | **0.242** | **0.282** | **4.576** | 0.257 | 0.290 | 4.871 | 0.251 | 0.284 | 4.605 |
> > > | Wind | **0.977** | **0.688** | **6.457** | 1.018 | 0.717 | 6.728 | 0.998 | 0.706 | 6.628 |
> > >
> > > **Note on MASE values.** We note that MASE values greater than 1 are expected in our setting. Specifically, we use a seasonal period of \( m = 1 \), while all forecasting models perform direct multi-step prediction (e.g., 96-step ahead) in a one-shot manner. Under this protocol, only the first prediction step has access to the most recent ground-truth observation, whereas subsequent steps do not have access to intermediate lag-1 values. As a result, the naive seasonal-1 benchmark (which relies on true lag-1 observations at each step) is relatively strong, making MASE > 1 a natural outcome. Therefore, MASE values in this range should be interpreted comparatively across methods rather than in absolute terms.
> > >
> > > **Plug-and-play results on Wind, AQShunyi (AQS), and AQWan (AQW), using a look-back window of 720 and computing MASE with seasonal period m=96.**
> > >
> > > | Dataset | Wind |  |  |  |  | AQS |  |  |  |  | AQW |  |  |  |
> > > | :-- | :--: | :--: | :--: | :--: | :--: | :--: | :--: | :--: | :--: | :--: | :--: | :--: | :--: | :--: |
> > > | **Horizon** | **96** | **192** | **336** | **720** | \| | **96** | **192** | **336** | **720** | \| | **96** | **192** | **336** | **720** |
> > > | | MASE | MASE | MASE | MASE | \| | MASE | MASE | MASE | MASE | \| | MASE | MASE | MASE | MASE |
> > > | iTransformer | 0.747 | 0.836 | 0.896 | 0.931 | \| | 0.766 | 0.797 | 0.817 | 0.875 | \| | 0.780 | 0.806 | 0.833 | 0.888 |
> > > | +PULSE | **0.689** | **0.788** | **0.846** | **0.900** | \| | **0.731** | **0.770** | **0.781** | **0.813** | \| | **0.736** | **0.764** | **0.821** | **0.832** |
> > > | DLinear | 0.750 | 0.809 | 0.861 | **0.894** | \| | 0.752 | 0.789 | 0.820 | 0.911 | \| | 0.769 | 0.809 | 0.833 | 0.901 |
> > > | +PULSE | **0.716** | **0.778** | **0.842** | 0.895 | \| | **0.731** | **0.768** | **0.797** | **0.863** | \| | **0.730** | **0.768** | **0.817** | **0.875** |
> > > | TimesNet | 0.793 | 0.914 | 0.982 | 1.062 | \| | 0.755 | 0.790 | 0.802 | **0.842** | \| | 0.839 | 0.861 | 0.824 | **0.822** |
> > > | +PULSE | **0.729** | **0.802** | **0.849** | **0.919** | \| | **0.734** | **0.787** | **0.799** | 0.849 | \| | **0.736** | **0.792** | **0.803** | 0.850 |
> > > | PatchTST | 0.760 | 0.848 | 0.882 | 0.924 | \| | 0.762 | 0.802 | 0.821 | 0.855 | \| | 0.769 | 0.805 | 0.819 | 0.893 |
> > > | +PULSE | **0.691** | **0.795** | **0.860** | **0.904** | \| | **0.741** | **0.782** | **0.799** | **0.852** | \| | **0.743** | **0.776** | **0.802** | **0.862** |
> > >
> > > [1] TFB: Towards Comprehensive and Fair Benchmarking of Time Series Forecasting Methods. Proceedings of the VLDB Endowment (PVLDB), 2024.
> > >
> > > [2] LargeST: A Benchmark Dataset for Large-Scale Traffic Forecasting. Advances in Neural Information Processing Systems (NeurIPS), 2023.

---

### Official Review · Reviewer_MCnk · 2026-03-12

**Soundness:** 3
**Presentation:** 3
**Significance:** 3
**Originality:** 4
**Overall Recommendation:** 4
**Confidence:** 4

**Summary:**

This paper presents PULSE, a framework addressing "Phase Amnesia" in non-stationary multivariate time series forecasting. Grounded in three physical hypotheses (Wold Decomposition, Dynamical Phase Evolution, and Heteroscedastic Residual Manifold), PULSE features a "Disentangle-Evolve-Simulate" pipeline. It explicitly decouples deterministic phase anchors from stochastic residuals, dynamically evolving the anchors via a Generative Phase Router while regularizing the residuals using a Statistic-Aware Mixup (SAM). Empirically, equipping a simple MLP with PULSE yields state-of-the-art (SOTA) results across 12 real-world benchmarks and demonstrates robust plug-and-play efficacy.

**Compliance With Llm Reviewing Policy:**

Affirmed.

**Key Questions For Authors:**

* Q1. Given the assumption of uncorrelated residuals in Remark 3.3 used to justify linear energy addition, how does the SAM module avoid theoretical fragility and remain robust when applied to real-world datasets that exhibit strong localized volatility clustering (GARCH effects)?
* Q2. Please respond to the theoretical and comparative comments raised in the Weaknesses section.

**Limitations:**

Yes

**Strengths And Weaknesses:**

**Strengths:**

* S1. Theoretical Rigorousness: The formal mathematical proofs detailing the limitations of existing frameworks (e.g., gradient attenuation) provide a highly grounded motivation for the proposed architecture.
* S2. Architectural Decoupling: The shift from passive structural modeling to generative evolution is conceptually highly innovative. Disentangling structural drift from stochastic noise successfully mirrors physical system dynamics.
* S3. Universal Applicability: The consistent performance gains achieved when integrating PULSE into disparate architectural backbones validate the fundamental necessity of the disentanglement strategy.

**Weaknesses:**

* W1. Asymmetric Representation Learning: Figures 3 and 9 show the historical anchor $A_{x}$ as highly volatile, while the generated future anchor $A_{y}$ is notably smooth. This asymmetry is visually striking but lacks a brief theoretical explanation in the main text regarding why the model suppresses historical volatility in the future state.
* W2. Frequency-Domain Loss Justification: Equation 8 adopts a pure Frequency-Domain MAE. While the manuscript briefly mentions that this prevents the "washing out" of high-frequency components, the exact functional role of this specific loss within the broader *Disentangle-Evolve-Simulate* paradigm isn't entirely clear to me. For instance, concurrent work such as FreDF (Wang et al., 2025a) explicitly advocates for a dual-domain loss setup. Clarifying why a pure frequency loss is chosen here, and conceptually how it differs from FreDF's dual-domain approach, would further strengthen the methodological narrative.
* W3. Algorithmic Distinction: The manuscript distinguishes itself from passive structure-based models, but a concise comparison of PULSE's decomposition sequence versus recent explicit decomposition baselines (like CycleNet or TimeEmb) is needed to clarify the core algorithmic novelty.

---

> ### Author Rebuttal · Authors · 2026-03-26
>
> We sincerely thank the reviewer for the insightful and technically rigorous comments. Below, we respond to the questions on representation asymmetry, the frequency-domain objective, the distinction from decomposition baselines, and SAM's robustness under volatility clustering.
>
> **W1: Representation Asymmetry**
>
> As discussed in Appendix B.5, the asymmetry between $A_x$ and $A_y$ is intrinsic to our design. Per your suggestion, we will clarify this explanation in the main text.
>
> - **$A_x$ as a structural absorber.** $A_x$ remains relatively volatile because it is designed to absorb the dominant high-energy structural variations from the raw signal. If $A_x$ were overly smooth, structural dynamics would leak into the residual $R_x$, weakening disentanglement and interfering with residual modeling.
> - **$A_y$ as condensed future structure.** In contrast, $A_y$ is designed to represent the evolved future structure. Its smoothness is therefore consistent with the low-rank hypothesis. Rather than directly copying historical fluctuations, the Generative Phase Router evolves the deterministic structural anchor under residual context, producing a smoother future anchor while leaving stochastic variation to the residual branch.
>
> **W2: Role of Frequency-Domain Loss and the Difference from FreDF**
>
> The frequency-domain MAE is introduced specifically to make Statistic-Aware Mixup (SAM) well-behaved on stochastic residuals. Direct interpolation in the time domain can suffer from **phase cancellation**. For example, mixing $R_1(t)=\sin(\omega t)$ and $R_2(t)=\sin(\omega t+\pi)$ yields a near-zero mixed target, even though both signals contain the same spectral energy. Under a time-domain objective, this cancellation encourages amplitude shrinkage and leads to scale collapse.
>
> By optimizing in the frequency domain, the model is guided to preserve the **spectral energy distribution** of the mixed residuals, which remains stable under such phase mismatch. This is why we adopt a pure frequency-domain objective in PULSE.
>
> This also differs conceptually from FreDF. FreDF emphasizes balancing supervision in both time and frequency domains through a dual-domain loss design. In contrast, our objective is tied to the training dynamics of SAM: it stabilizes residual learning under statistic-aware interpolation, rather than balancing two views of the same signal. In this sense, the frequency-domain loss in PULSE is not a generic replacement of time-domain supervision, but a targeted mechanism working alongside disentanglement and SAM.
>
> **W3: Algorithmic Distinction from Explicit Decomposition Baselines**
>
> Our method differs from recent explicit decomposition baselines like CycleNet and TimeEmb in two key aspects.
>
> - **Decompose first, then normalize.** Existing methods typically normalize the whole signal before decomposition. As analyzed in Proposition 3.1, this may mask the scale of subtle residual variations by the global standard deviation. PULSE instead isolates the stochastic residual first and applies normalization only to that component, preserving gradient sensitivity for small but important fluctuations.
> - **Generate future structure instead of reusing historical structure.** Existing structure-based methods generally assume that historical periodicity acts as a stable prior for the future. PULSE avoids this static assumption. After disentanglement, the anchor and residual are fed into the Phase Router to generate dynamically evolving future phase structure. This changes the role of decomposition from passive historical lookup to active future structure simulation.
>
> **Q1: Assumption of Uncorrelated Residuals and Volatility Clustering (e.g., GARCH effects)**
>
> We appreciate this important question. The key distinction is between **intra-sequence temporal dependence** and **inter-sample independence during mixup**. The assumption in Remark 3.3 applies strictly to the latter: it concerns two randomly paired residual sequences $R_i$ and $R_j$ sampled from different contexts in a batch, rather than independence across timesteps within a single sequence. Therefore, our formulation does not deny the presence of volatility clustering in real-world series.
>
> In fact, the framework is designed precisely for heteroscedastic residual dynamics. Our phase-anchored disentanglement isolates the stochastic residual manifold from the deterministic structural anchor, while SAM interpolates residual statistics from different contexts to expose the model to a broader range of volatility patterns. In this sense, the residual stream may be less locally smooth in the time domain, while still exhibiting structured heteroscedastic statistics at the distribution level. Thus, SAM should be understood as a regularization strategy motivated by heteroscedastic residual variation, rather than a claim that real residual processes are temporally uncorrelated.

---

> > ### Author Rebuttal · Reviewer_MCnk · 2026-04-02
> >
> > Thank you for your detailed response. I will keep my original score.

---

> > > ### Author Response · Authors · 2026-04-02
> > >
> > > We sincerely thank the reviewer  for carefully reading our rebuttal. We appreciate the confirmation that our clarifications regarding representation asymmetry, the frequency-domain objective, decomposition distinctions, and SAM’s behavior under heteroscedastic residuals addressed the concerns.
> > >
> > > We note that the reviewer intends to keep their original score, and we respect this decision.
> > >
> > > Thank you again for your time and thoughtful consideration.

---

### Decision · Program_Chairs · 2026-04-30

**Decision:**

Accept (regular)

**Comment:**

This paper presents an framework for non-stationary multivariate time series forecasting, with a conceptually appealing disentangle-evolve-simulate design and strong empirical results in the reported evaluation setting. The reviewers found the core idea promising, particularly the explicit decoupling of deterministic phase dynamics and stochastic residuals, as well as the method’s parameter efficiency and plug-and-play benefits across backbones. At the same time, there are concerns about the current empirical scope, especially the reliance on short context lengths, the limited validation of the claimed “Phase Amnesia” failure mode, and the need for clearer methodological exposition and broader evaluation. Overall, the paper appears promising and potentially impactful, but there are still remaining questions about generality and experimental completeness.